# Exploring the Design Space of Diffusion Bridge Models via Stochasticity Control

## Abstract

Diffusion bridge models effectively facilitate image-to-image (I2I) translation by connecting two distributions. However, existing methods overlook the impact of noise in sampling SDEs, transition kernel, and the base distribution on sampling efficiency, image quality and diversity. To address this gap, we propose the Stochasticity-controlled Diffusion Bridge (SDB), a novel theoretical framework that extends the design space of diffusion bridges, and provides strategies to mitigate singularities during both training and sampling. By controlling stochasticity in the sampling SDEs, our sampler achieves speeds up to $5\times$ faster than the baseline, while also producing lower FID scores. After training, SDB sets new benchmarks in image quality and sampling efficiency via managing stochasticity within the transition kernel. Furthermore, introducing stochasticity into the base distribution significantly improves image diversity, as quantified by a newly introduced metric. Code would be available on Github repo.

## 1 Introduction

Denoising Diffusion Models (DDMs) create a stochastic process to transition Gaussian noise into a target distribution (Song & Ermon, 2019; Ho et al., 2020; Song et al., 2020). Building upon this, diffusion bridge-based models (DBMs) have been developed to transport between two arbitrary distributions, $\pi_T$ and $\pi_0$, including Bridge Matching (Peluchetti, 2023), Flow Matching (Lipman et al., 2022), and Stochastic Interpolants (Albergo et al., 2023). Compared to DDMs, DBMs offer greater versatility for tasks such as I2I translation (Linqi Zhou et al., 2023; Liu et al., 2023). This advantage arises because using a Gaussian prior often fails to incorporate sufficient knowledge about the target distribution.

In general, there are two primary design philosophies for DBMs. The first involves deriving a pinned process (Yifeng Shi et al., 2023) from a given reference process (e.g., Brownian motion) via Doob's $h$-transform, and then constructing a bridge to approach it (Linqi Zhou et al., 2023; Peluchetti, 2023). The second regime aims to directly design a bridge based on a specified transition kernel (Lipman et al., 2022; Albergo et al., 2023). While the former also results in a transition kernel, the mean and variance in the kernel are *coupled*, which limits the design flexibility for possible bridges. In this work, we follow the second fashion and further propose the Stochasticity Control (SC) mechanism, which facilitates easier tuning and leads to enhanced performance across a variety of tasks. Our main contributions are as follows:

- We introduce the **Stochasticity-controlled Diffusion Bridge** (SDB), a generalized framework that adopts a transition kernel-based design philosophy to elucidate the design space of DBMs, shown in Fig. 10. Notably, this framework not only encompasses other mainstream DBMs such as DDBM (Linqi Zhou et al., 2023) and I2SB (Liu et al., 2023), but also DDMs like EDM (Karras et al., 2022), as detailed in Table 1.

- A **Stochasticity Control** (SC) mechanism is proposed by adding noise into the base distribution, designing a noise schedule for the transition kernel, and regulating the drift term in the sampling SDEs. In addition, we explore **score reparameterization** and the **discretization schemes** of sampling SDEs to mitigate singularity during training and sampling. These combined strategies lead to significant improvements in training stability, sampling efficiency, output quality, and conditional diversity.

Figure 1: An illustration of the framework for constructing diffusion bridge models. The parameters $b$, $\gamma_t$, and $\epsilon_t$ govern the stochasticity introduced at three main stages: preprocessing, training, and sampling. Specifically, $b$ determines the noise added to the base distribution during preprocessing, $\gamma_t$ controls the noise introduced into the transition kernel, impacting both training and sampling, and $\epsilon_t$ regulates the noise added to the sampling SDEs, affecting only the sampling stage.

- Experimental results show that our sampler operates $5\times$ faster than the DDBM sampler and achieves a lower FID score using the same pretrained models. When trained from scratch, our model sets a new benchmark for image quality, requiring only 5 function evaluations to reach an FID of 0.89 on Edges2handbags ($64 \times 64$) and 4.16 on DIODE ($256 \times 256$) datasets. Furthermore, by introducing noise into the base distribution, we significantly enhance the diversity of synthetic images, resulting in a greater variety of colors and textures.

**Notations** Let $\pi_T$, $\pi_0$, and $\pi_{0T}$ represent the base distribution, the target distribution, and the joint distribution of them respectively. $\pi_{\text{cond}}$ and $\pi_{\text{data}}$ represent the distributions of the input and output data. Let $p$ be the distribution of a diffusion process; we denote its marginal distribution at time $t$ by $p_t$, the conditional distribution at time $t$ given the state at time $s$ by $p_{t|s}$, and the distribution at time $t$ given the states at times 0 and $T$ by $p_{t|0T}$, i.e., the transition kernel of a bridge.

## 2 BACKGROUND

### 2.1 DENOISING DIFFUSION MODELS

Denoising diffusion models map target distribution $\pi_0$ into a base distribution $\pi_T$ by define a forward process on the time-interval $[0, T]$:

$$d\mathbf{X}_t = \bar{f}_t \mathbf{X}_t dt + \bar{g}_t d\mathbf{W}_t, \quad \mathbf{X}_0 \sim \pi_0, \tag{1}$$

where $\bar{f}_t, \bar{g}_t : [0, T] \to \mathbb{R}$ is the scalar-valued drift and diffusion term, $\mathbf{X}_0 \in \mathbb{R}^d$ is drawn from the target distribution $\pi_0$, $\mathbf{W}_t$ is a $d$-dimensional Wiener process. To sample from the target distribution $\pi_0$, the generative model is given by the reverse SDE or ODE (Song et al., 2020):

$$d\mathbf{X}_t = \left[ \bar{f}_t \mathbf{X}_t - \bar{g}_t^2 \nabla_{\mathbf{X}_t} \log q_t(\mathbf{X}_t) \right] dt + \bar{g}_t d\mathbf{W}_t, \quad \mathbf{X}_T \sim \pi_T, \tag{2}$$

$$d\mathbf{X}_t = \left[ \bar{f}_t \mathbf{X}_t - \frac{1}{2} \bar{g}_t^2 \nabla_{\mathbf{X}_t} \log q_t(\mathbf{X}_t) \right] dt, \quad \mathbf{X}_T \sim \pi_T, \tag{3}$$

where $q_t$ denotes the marginal distribution of this process. The score function $\nabla_{\mathbf{x}_t} \log q_t(\mathbf{x}_t)$ is approximated using a neural network $\mathbf{s}_\theta(\mathbf{x}_t, t)$, which can be learned by the score-matching loss:

$$\mathcal{L}(\theta) = \mathbb{E}_{\mathbf{x}_t \sim p_{t|0}(\mathbf{x}_t|\mathbf{x}_0), \mathbf{x}_0 \sim \pi_0, t \sim \mathcal{U}(0,T)} \left[ \omega(t) \left\| \mathbf{s}_\theta(\mathbf{x}_t, t) - \nabla_{\mathbf{x}_t} \log q_{t|0}(\mathbf{x}_t|\mathbf{x}_0) \right\|^2 \right], \tag{4}$$

where $q_{t|0}$ is the analytic forward transition kernel and $\omega(t)$ is a positive weighting function.

## 2.2 DENOISING DIFFUSION BRIDGE MODELS

DDBMs (Linqi Zhou et al., 2023) extend diffusion models to translate between two arbitrary distributions $\pi_0$ and $\pi_T$ given samples from them. Consider a reference process in Eq. (1) with transition kernel $q_{t|0}(\mathbf{x}_t|\mathbf{x}_0) = \mathcal{N}(\mathbf{x}_t; a_t\mathbf{x}_0, \sigma_t^2\mathbf{I})$, this process can be pinned down at an initial and terminal point $\mathbf{x}_0, \mathbf{x}_T$. Under mild assumptions, the pinned process is given by Doob's $h$-transform (Rogers & Williams, 2000):

$$d\mathbf{X}_t = \{\bar{f}_t\mathbf{X}_t + \bar{g}_t^2\nabla_{\mathbf{X}_t}\log p_{T|t}(\mathbf{x}_T|\mathbf{X}_t)\}dt + \bar{g}_t d\mathbf{W}_t, \quad \mathbf{X}_0 = \mathbf{x}_0, \tag{5}$$

where $\nabla_{\mathbf{X}_t}\log p_{T|t}(\mathbf{x}_T \mid \mathbf{X}_t) = \frac{(a_t/a_T)\mathbf{x}_T - \mathbf{X}_t}{\sigma_t^2(\mathrm{SNR}_t/\mathrm{SNR}_T - 1)}$ and $\mathrm{SNR} := a_t^2/\sigma_t^2$ (Linqi Zhou et al., 2023). The marginal density of process (5) serves as transition kernel and is given by $p(\mathbf{x}_t|\mathbf{x}_0, \mathbf{x}_T) = \mathcal{N}(\mathbf{x}_t; \alpha_t\mathbf{x}_0 + \beta_t\mathbf{x}_T, \gamma_t^2\mathbf{I})$, where $\alpha_t = a_t(1 - \frac{\mathrm{SNR}_T}{\mathrm{SNR}_t})$, $\beta_t = \frac{a_t}{a_T}\frac{\mathrm{SNR}_T}{\mathrm{SNR}_t}$, $\gamma_t^2 = \sigma_t^2(1 - \frac{\mathrm{SNR}_T}{\mathrm{SNR}_t})$.

To sample from the conditional distribution $p(\mathbf{x}_0|\mathbf{x}_T)$, we can solve the reverse SDE or probability flow ODE from $t = T$ to $t = 0$:

$$d\mathbf{X}_t = \{\bar{f}_t\mathbf{X}_t + \bar{g}_t^2(\nabla_{\mathbf{X}_t}\log p_{T|t}(\mathbf{x}_T|\mathbf{X}_t) - \nabla_{\mathbf{X}_t}\log p_{t|T}(\mathbf{X}_t|\mathbf{x}_T))\}dt + \bar{g}_t d\mathbf{W}_t, \mathbf{X}_T = \mathbf{x}_T \tag{6}$$

$$d\mathbf{X}_t = \{\bar{f}_t\mathbf{X}_t + \bar{g}_t^2(\nabla_{\mathbf{X}_t}\log p_{T|t}(\mathbf{x}_T|\mathbf{X}_t) - \frac{1}{2}\nabla_{\mathbf{X}_t}\log p_{t|T}(\mathbf{X}_t|\mathbf{x}_T))\}dt, \quad \mathbf{X}_T = \mathbf{x}_T. \tag{7}$$

Generally, the score $\nabla_{\mathbf{x}_t}\log p_{t|T}(\mathbf{x}_t|\mathbf{x}_T)$ in Eqs. (6) and (7) is intractable. However, it can be effectively estimated by denoising bridge score matching. Let $(\mathbf{x}_0, \mathbf{x}_T) \sim \pi_{0,T}(\mathbf{x}_0, \mathbf{x}_T)$, $\mathbf{x}_t \sim p_{t|0,T}(\mathbf{x}_t|\mathbf{x}_0, \mathbf{x}_T)$, $t \sim \mathcal{U}(0, T)$, and $\omega(t)$ be non-zero loss weighting term of any choice, then the score $\nabla_{\mathbf{x}_t}\log p_{T|t}(\mathbf{x}_T|\mathbf{x}_t)$ can be approximated by a neural network $\mathbf{s}_\theta(\mathbf{x}_t, \mathbf{x}_T, t)$ with denoising bridge score matching objective:

$$\mathcal{L}(\theta) = \mathbb{E}_{\mathbf{x}_t, \mathbf{x}_0, \mathbf{x}_T, t}\left[w(t)\|\mathbf{s}_\theta(\mathbf{X}_t, \mathbf{x}_T, t) - \nabla_{\mathbf{x}_t}\log p_{t|0,T}(\mathbf{X}_t \mid \mathbf{x}_0, \mathbf{x}_T)\|^2\right]. \tag{8}$$

To sum up, DDBM starts with the forward SDE outlined in Eq. (1) with a marginal distribution of $q_{t|0}(\mathbf{x}_t|\mathbf{x}_0) = \mathcal{N}(\mathbf{x}_t; a_t\mathbf{x}_0, \sigma_t^2\mathbf{I})$. The pinned process is then built by applying Doob's $h$-transform as specified in Eq. (5), which is unnecessarily complicated and constraining. Additionally, the transition kernel of the pinned process becomes complex and coupled, as $\alpha_t$, $\beta_t$, and $\gamma_t$ are all interrelated through $a_t$ and $\sigma_t$, increasing the design difficulty. In the next section, we will demonstrate how $\alpha_t$ and $\beta_t$ can be used to control interpolation, while $\gamma_t$ is designed to regulate the stochasticity introduced into the path.

## 3 STOCHASTICITY CONTROL

### 3.1 STOCHASTICITY CONTROL IN TRANSITION KERNEL

We are interested in building a diffusion process to transport from two arbitrary distributions $\pi_T$ and $\pi_0$. Suppose the transition kernel of this process is $p_{t|0,T}(\mathbf{x}_t|\mathbf{x}_0, \mathbf{x}_T) = \mathcal{N}(\mathbf{x}_t; \alpha_t\mathbf{x}_0 + \beta_t\mathbf{x}_T, \gamma_t^2\mathbf{I})$. For diffusion models, we can simply let $\beta_t = 0$ and $\alpha_0 = 1$ and $\gamma_0 = 0$. For bridge models, to ensure that the process originates from $\mathbf{x}_0$ and concludes at $\mathbf{x}_T$, we set $\alpha_0 = \beta_T = 1$ and $\alpha_T = \beta_0 = 0$. Additionally, we require $\alpha_t, \beta_t, \gamma_t > 0$ for $t \in (0, T)$. Let $T = 1$, one simple design example involves defining $\alpha_t$ and $\beta_t$ linearly, such that $\alpha_t = 1 - t$ and $\beta_t = t$, with $\gamma_t = 2\gamma_{\max}\sqrt{t(1-t)}$, where $\gamma_{\max}$ is a constant representing the maximum noise level. This configuration is referred to as the *linear path for transition kernel*. Other designs such as $\alpha_t = \cos(\pi t/2)$, $\beta_t = \sin(\pi t/2)$, and $\gamma_t = \sin(\pi t)$ can also be employed. Notably, the DDBM-VP and DDBM-VE models presented in (Linqi Zhou et al., 2023) can be considered as special cases within our framework, contingent upon the specific choices of $\alpha_t$, $\beta_t$, and $\gamma_t$, see Table 1 and Appendix C for more details. In this paper, we limit our scope on Linear transition kernel, i.e., $p_{t|0,T}(\mathbf{x}_t|\mathbf{x}_0, \mathbf{x}_T) = \mathcal{N}(\mathbf{x}_t; (1-t)\mathbf{x}_0 + t\mathbf{x}_0, 4\gamma_{\max}^2 t(1-t)\mathbf{I})$, A detailed discussion on the rationale behind the choices of $\alpha_t$, $\beta_t$, and and an ablation study on the shape of $\gamma_t$ is provided in D.

Table 1: Specify design choices for different model families. In the implementation, $\sigma_t = t$ for EDM, $\sigma_t = t, a_t = 1$ for DDBM-VE, $\sigma_t = \sqrt{e^{\frac{1}{2}\beta_d t^2 + \beta_{\min} t} - 1}$ and $a_t = 1/\sqrt{e^{\frac{1}{2}\beta_d t^2 + \beta_{\min} t}}$ for DDBM-VP, where $\beta_d$ and $\beta_{\min}$ are parameters. We include details and proofs in Appendix C.

| | | I2SB | DDBM | EDM | Ours |
|---|---|---|---|---|---|
| SC-transition kernel Sec. 3.1 | $\alpha_t$ | $1 - \sigma_t^2/\sigma_T^2$ | $a_t(1 - a_T^2\sigma_t^2/(\sigma_T^2 a_t^2))$ | $1$ | $1 - t$ |
| | $\beta_t$ | $\sigma_t^2/\sigma_T^2$ | $a_T\sigma_t^2/(\sigma_T^2 a_t)$ | $0$ | $t$ |
| | $\gamma_t^2$ | $\sigma_t^2(1 - \sigma_t^2/\sigma_T^2)$ | $\sigma_t^2(1 - a_T^2\sigma_t^2/(\sigma_T^2 a_t^2))$ | $\sigma_t^2$ | $\frac{\gamma_{\max}^2}{4}t(1-t)$ |
| SC-sampling SDEs Sec. 3.2 | $\epsilon_t$ | $\frac{\gamma_{t-\Delta t}^2\beta_t^2 - \beta_{t-\Delta t}^2\gamma_t^2}{2\beta_t^2\Delta t}$ | $\eta(\gamma_t\dot{\gamma}_t - \frac{\dot{\alpha}_t}{\alpha_t}\gamma_t^2)$ $\eta = 0$ or $\eta = 1$ | $\bar{\beta}_t\sigma_t^2$ - | $\eta(\gamma_t\dot{\gamma}_t - \frac{\dot{\alpha}_t}{\alpha_t}\gamma_t^2)$ $\eta \in [0,1]$ |
| SC-base distribution Sec. 3.3 | $\pi_T$ | $\pi_{\text{cond}}$ | $\pi_{\text{cond}}$ | $\pi_{\text{cond}}$ | $\pi_{\text{cond}} * \mathcal{N}(0, b^2\mathbf{I})$ |
| Score reparameterization Sec. 4.1 | $\mathbf{s}_\theta$ | $\frac{\alpha_t(x_t - \hat{\epsilon}\sigma_t) + \beta_t\mathbf{x}_T - \mathbf{x}_t}{\gamma_t^2}$ | $\frac{\alpha_t\hat{\mathbf{x}}_0 + \beta_t\mathbf{x}_T - \mathbf{x}_t}{\gamma_t^2}$ | $\frac{\alpha_t\hat{\mathbf{x}}_0 + \beta_t\mathbf{x}_T - \mathbf{x}_t}{\gamma_t^2}$ | $\frac{\alpha_t\hat{\mathbf{x}}_0 + \beta_t\mathbf{x}_T - \mathbf{x}_t}{\gamma_t^2}$ |
| Discretization Sec. 4.2 | - | Euler Eq. (17) | Euler Eq. (14) | Heun - | Euler Eqs. (14) and (16) |

## 3.2 STOCHASTICITY CONTROL IN SAMPLING SDES

Stochasticity control (SC) during the sampling phase has been explored for diffusion models by Karras et al. (2022), yet comprehensive studies on its application to diffusion bridge models remain limited. Eqs. (19) and (20) offer sampling schemes that align with Eqs. (6) and (7) in the DDBM framework. However, these methods do not guarantee optimal performance in terms of sampling speed and image quality. To address this issue, Linqi Zhou et al. (2023) introduced a hybrid sampler alternating between reversed ODE and SDE, and Zheng et al. (2024) accelerated sampling with an improved algorithm using discretized timesteps. This section aims to explore how SC can be further optimized in the sampling for DBMs, thereby addressing the current research gap. Given transition kernel, we can identify the reverse sampling SDEs, as demonstrated in Theorem 1.

**Theorem 1.** *Suppose the transition kernel of a diffusion process is given by $p_{t|0,T}(\mathbf{x}_t \mid \mathbf{x}_0, \mathbf{x}_T) = \mathcal{N}(\mathbf{x}_t; \alpha_t\mathbf{x}_0 + \beta_t\mathbf{x}_T, \gamma_t^2\mathbf{I})$, then the evolution of conditional probability $q(\mathbf{X}_t|\mathbf{x}_T)$ has a class of time reverse sampling SDEs of the form:*

$$d\mathbf{X}_t = \left[\dot{\alpha}_t\hat{\mathbf{x}}_0 + \dot{\beta}_t\mathbf{x}_T - (\dot{\gamma}_t\gamma_t + \epsilon_t)\nabla_{\mathbf{X}_t}\log p_t(\mathbf{X}_t|\mathbf{x}_T)\right]dt + \sqrt{2\epsilon_t}d\mathbf{W}_t \quad \mathbf{X}_T = \mathbf{x}_T. \quad (9)$$

**Remark 3.1.** *As $\epsilon_t = 0$, Eq. (9) recovers the sampling ODE specified in Eq. (7). As $\epsilon_t = \gamma_t\dot{\gamma}_t - \frac{\dot{\alpha}_t}{\alpha_t}\gamma_t^2$, Eq. (9) recovers the sampling SDE specified in Eq. (6). As $\epsilon_t = \eta(\gamma_t\dot{\gamma}_t - \frac{\dot{\alpha}_t}{\alpha_t}\gamma_t^2), \eta \in (0,1)$, the stochasticity is between the original sampling ODE in Eq. (7) and SDE. in Eq. (6).*

There is no definitive principle for designing $\epsilon_t$. For DDMs, Karras et al. (2022) suggest that the optimal level of stochasticity should be determined empirically. In the case of DBMs, however, certain design guidelines can be followed to potentially enhance performance. Unlike DDMs, which typically start sampling from Gaussian noise, DBMs begin with a deterministic condition $\mathbf{x}_T$. Therefore, setting $\epsilon_t = 0$ results in no stochasticity for the sampling process and final sample $\mathbf{x}_0$, which may partly explain the poor performance of ODE samplers in this context. However, it is advantageous to set $\epsilon_t = 0$ during the final steps of sampling. The rationale behind this approach is discussed in detail in Section 4.2.

## 3.3 STOCHASTICITY CONTROL IN BASE DISTRIBUTION

Conditional diversity refers to the range of outputs that can be generated from specific conditions. This is valuable in scenarios like image generation from edges, where one edge image may lead to multiple valid images differing in color, texture, or detail. Conversely, in super-resolution, where

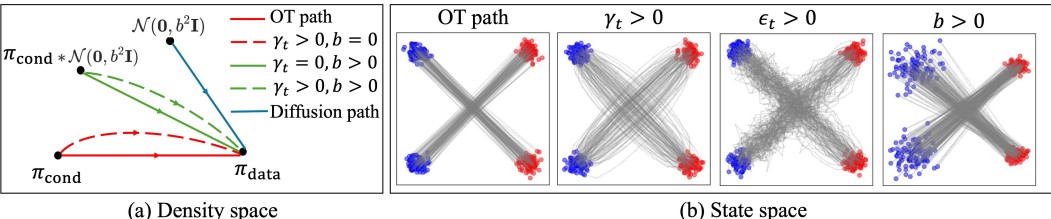

(a) Density space                                    (b) State space

Figure 2: The effect of stochasticity control on density and state spaces. Adding no stochasticity ($\gamma_t = 0, \epsilon_t = 0, b = 0$) leads to the optimal transport (OT) path. (a). In the density space, OT path directly links $\pi_{\text{cond}}$ and $\pi_{\text{data}}$, while diffusion path transports from $\mathcal{N}(0, b^2\mathbf{I})$ to $\pi_{\text{data}}$. When $\gamma_t > 0$ (dash lines), it increases stochasticity in the middle of the transition, whereas $b > 0$ (green lines), it directly adds stochasticity to the base distribution, leading to trade off between DDMs and DBMs when $b = 0$. (b). In the state space, we use blue dots and red dots to represent input and output data respectively. The OT path directly links two samples, it shows a detoured path when $\gamma_t > 0$, introduces a zigzag pattern while $\epsilon_t > 0$, and smooths the base distribution as $b > 0$.

a high-resolution image is created from a low-resolution one, output variability is limited by the input's structure, demanding consistency and fidelity to the original rather than diversity.

To control the conditional diversity of diffusion bridge models, we can trade off between DBMs and DDMs by controlling the stochasticity in the base distribution. Bridge models transport the base distribution $\pi_T$ to target distribution $\pi_0$. Typically, most previous bridge models, such as those discussed in (Linqi Zhou et al., 2023; Albergo et al., 2023), treat $\pi_T$ as the input data distribution, $\pi_{\text{cond}}$. However, it is flexible to design $\pi_T$; for instance, by choosing $\pi_T$ as a Gaussian distribution, we recover DDMs. An intermediate approach involves the convolution of $\pi_{\text{cond}}$ with a Gaussian distribution, $\pi_T = \pi_{\text{cond}} * \mathcal{N}(0, b^2\mathbf{I})$, where $b$ is a constant that controls the strength of booting noise we added to the input data distribution. We provide an illustration of the effect of SC in transition kernel, sampling SDEs and distribution in Fig. 2.

We developed the Average Feature Distance (AFD) metric to quantify the conditional diversity among generated images. Initially, we select a group of source images $\{\mathbf{x}_T^{(i)}\}_{i=1}^M$. For each $\mathbf{x}_T^{(i)}$, we then generate $L$ distinct target samples. The $j$-th generated sample corresponding to the $i$-th source image is denoted by $\mathbf{y}_{ij}$. Then the AFD is calculated as follows:

$$\text{AFD} = \frac{1}{M} \sum_{i=1}^M \frac{1}{L^2 - L} \sum_{k,l=1, k \neq l}^L \|F(\mathbf{y}_{ik}) - F(\mathbf{y}_{il})\| \tag{10}$$

where $F(\cdot)$ is a function that extracts the features of images, and $\|\cdot\|$ represents Euclidean norm. Intuitively, a larger AFD indicates the better conditional diversity. Here, $F(\mathbf{x})$ can be $\mathbf{x}$ to evaluate the diversity directly in the pixel space. Alternatively, $F(\cdot)$ can be defined using the Inception-V3 model to assess the diversity in the latent space. In our experiments, we use AFD in latent space.

## 4 SCORE REPARAMETERIZATION AND ALGORITHM DESIGN

### 4.1 SCORE REPARAMETERIZATION

The log gradient of Gaussian transition kernel $p_{t|0,T}(\mathbf{x}_t|\mathbf{x}_0, \mathbf{x}_T) = \mathcal{N}(\mathbf{x}_t; \alpha_t\mathbf{x}_0 + \beta_t\mathbf{x}_T, \gamma_t^2\mathbf{I})$ has an analytical form: $\nabla_{\mathbf{x}_t} \log p_{t|0,T}(\mathbf{X}_t \mid \mathbf{x}_0, \mathbf{x}_T) = (\alpha_t\mathbf{x}_0 + \beta_t\mathbf{x}_T - \mathbf{x}_t)/\gamma_t^2$. Therefore, the denoising bridge score matching objective in Eq. (8) is tractable. However, the singular term $1/\gamma_t^2$ at endpoints $t = 0$ and $t = T$ can lead to highly unstable training, see Appendix D for more details. Consequently, instead of directly parameterizing the score function $\nabla_{\mathbf{x}_t} \log p_t(\mathbf{x}_t|\mathbf{x}_T)$ with a neural network, we opt to reparameterize the score as a function of $\hat{\mathbf{x}}_0(\mathbf{x}_t, \mathbf{x}_T, t)$, as demonstrated in Theorem 2. This reparameterization strategy, initially introduced in EDM (Karras et al., 2022), is particularly significant for enhancing the stability and performance of our bridge models.

**Theorem 2.** *Let* $(\mathbf{x}_0, \mathbf{x}_T) \sim \pi_0(\mathbf{x}_0, \mathbf{x}_T)$, $\mathbf{x}_t \sim p_{t|0,T}(\mathbf{x}_t|\mathbf{x}_0, \mathbf{x}_T)$, $t \sim \mathcal{U}(0,T)$. *Given the transition kernel:* $p_{t|0,T}(\mathbf{x}_t \mid \mathbf{x}_0, \mathbf{x}_T) = \mathcal{N}\left(\mathbf{x}_t; \alpha_t \mathbf{x}_0 + \beta_t \mathbf{x}_T, \gamma_t^2 \mathbf{I}\right)$, *if* $\hat{\mathbf{x}}_0(\mathbf{x}_t, \mathbf{x}_T, t)$ *is a denoiser function that minimizes the expected* $L_2$ *denoising error for samples drawn from* $\pi_0(\mathbf{x}_0, \mathbf{x}_T)$*:*

$$\hat{\mathbf{x}}_0(\mathbf{x}_t, \mathbf{x}_T, t) = \arg\min_{D(\mathbf{x}_t, \mathbf{x}_T, t)} \mathbb{E}_{\mathbf{x}_0, \mathbf{x}_T, \mathbf{x}_t} \left[ \lambda(t) \| D(\mathbf{x}_t, \mathbf{x}_T, t) - \mathbf{x}_0 \|_2^2 \right], \tag{11}$$

*then the score has the following relationship with* $\hat{\mathbf{x}}_0(\mathbf{x}_t, \mathbf{x}_T, t)$*:*

$$\nabla_{\mathbf{x}_t} \log p_t(\mathbf{x}|\mathbf{x}_T) = \frac{\alpha_t \hat{\mathbf{x}}_0(\mathbf{x}, \mathbf{x}_T, t) + \beta_t \mathbf{x}_T - \mathbf{x}_t}{\gamma_t^2}. \tag{12}$$

The key observation is that $\hat{\mathbf{x}}_0(\mathbf{x}_t, \mathbf{x}_T, t)$ can be estimated by a neural network $D_\theta(\mathbf{x}_t, \mathbf{x}_T, t)$ trained according to Eq. (11). In the implementation, we include additional pre- and post-processing steps: scaling functions and loss weighting, see Appendix E for details.

## 4.2 ALGORITHM DESIGN

Let $\hat{\mathbf{z}}_t =: (\mathbf{x}_t - \alpha_t \hat{\mathbf{x}}_0 - \beta_t \mathbf{x}_T)/\gamma_t$, then the score $\nabla_{\mathbf{x}_t} \log p_t(\mathbf{x}|\mathbf{x}_T)$ and $\hat{\mathbf{z}}_t$ has a linear relationship: $\hat{\mathbf{z}}_t = -\gamma_t \nabla_{\mathbf{x}_t} \log p_t(\mathbf{x}|\mathbf{x}_T)$. An alternative formulation of the sampling SDEs (9) is presented as:

$$d\mathbf{X}_t = \left[ \dot{\alpha}_t \hat{\mathbf{x}}_0 + \dot{\beta}_t \mathbf{x}_T + (\dot{\gamma}_t + \frac{\epsilon_t}{\gamma_t})\hat{\mathbf{z}}_t \right] dt + \sqrt{2\epsilon_t} d\mathbf{W}_t. \tag{13}$$

Instead of using the score directly, we apply Eq. (13) to reduce truncation error. Additionally, $\hat{\mathbf{z}}$ can be seen as the estimated noise added to the interpolation (Albergo et al., 2023), the introduction of $\hat{\mathbf{z}}$ brings more interpretability. One discretization scheme of sampling SDEs Eq. (13) is based on Euler's method:

$$\mathbf{x}_{t-\Delta t} \approx \mathbf{x}_t - \left[ \dot{\alpha}_t \hat{\mathbf{x}}_0 + \dot{\beta}_t \mathbf{x}_T + (\dot{\gamma}_t + \frac{\epsilon_t}{\gamma_t})\hat{\mathbf{z}} \right] \Delta t + \sqrt{2\epsilon_t \Delta t} \bar{\mathbf{z}}_t, \quad \bar{\mathbf{z}}_t \sim \mathcal{N}(0, \mathbf{I}). \tag{14}$$

Furthermore, for small enough $\Delta t$ the derivative term can be approximated by: $\dot{\alpha}_t \approx (\alpha_t - \alpha_{t-\Delta t})/\Delta t$, $\dot{\beta}_t \approx (\beta_t - \beta_{t-\Delta t})/\Delta t$, $\dot{\gamma}_t \approx (\gamma_t - \gamma_{t-\Delta t})/\Delta t$. Using the fact that $\mathbf{x}_t = \alpha_t \hat{\mathbf{x}}_0 + \beta_t \mathbf{x}_T + \gamma_t \hat{\mathbf{z}}_t$, we can further simplify the iteration:

$$\mathbf{x}_{t-\Delta t} \approx \alpha_{t-\Delta t} \hat{\mathbf{x}}_0 + \beta_{t-\Delta t} \mathbf{x}_T + (\gamma_{t-\Delta t} - \frac{\epsilon_t \Delta t}{\gamma_t})\hat{\mathbf{z}}_t + \sqrt{2\epsilon_t \Delta t} \bar{\mathbf{z}}_t. \tag{15}$$

As $\gamma_{t-\Delta t}^2 - 2\epsilon_t \Delta t > 0$, $\gamma_{t-\Delta t} - \frac{\epsilon_t \Delta t}{\gamma_t} \approx \sqrt{\gamma_{t-\Delta t}^2 - 2\epsilon_t \Delta t}$, which leads to another discretization and recovers the sampler of DBIM (Zheng et al., 2024):

$$\mathbf{x}_{t-\Delta t} = \alpha_{t-\Delta t} \hat{\mathbf{x}}_0 + \beta_{t-\Delta t} \mathbf{x}_T + \sqrt{\gamma_{t-\Delta t}^2 - 2\epsilon_t \Delta t} \hat{\mathbf{z}}_t + \sqrt{2\epsilon_t \Delta t} \bar{\mathbf{z}}_t. \tag{16}$$

**Remark 4.1.** *Eq. (16) provides more insight about the noise and the design of* $\epsilon_t$*. Here* $\hat{\mathbf{z}}_t$ *and* $\bar{\mathbf{z}}_t$ *serve as predicted noise and added noise respectively. Generally, we assume the error* $\|\mathbf{x}_0 - \hat{\mathbf{x}}_0(\mathbf{x}_t, \mathbf{x}_T, t)\|$ *decreses as we move* $\mathbf{x}_t$ *from* $\mathbf{x}_T$ *to* $\mathbf{x}_0$*. Therefore, a small* $\epsilon_t$ *was suggested as* $t$ *close to 0. Further, due to the singular term* $\epsilon_t \Delta_t/\gamma_t$ *at* $t = 0$*, it's better to set* $\epsilon_t$ *small enough to avoid singularity.*

**Remark 4.2.** *Eq. (16) requires a constraint* $\gamma_{t-\Delta t}^2 - 2\epsilon_t \Delta t > 0$*. Note that this limitation is unnecessary and will limit the design of* $\epsilon_t$*.*

As $2\epsilon_t \Delta t = \gamma_{t-\Delta t}^2 - \beta_{t-\Delta t}^2 \gamma_t^2 / \beta_t^2$, the coefficient of $\mathbf{x}_t$ in Eq. 16 is 0, thus Eq. 16 can be simplified as:

$$\mathbf{x}_{t-\Delta t} = (\alpha_{t-\Delta t} - \alpha_t \frac{\beta_{t-\Delta t}}{\beta_t})\hat{\mathbf{x}}_0 + \frac{\beta_{t-\Delta t}}{\beta_t}\mathbf{x}_t + \sqrt{\gamma_{t-\Delta t}^2 - \frac{\beta_{t-\Delta t}^2 \gamma_t^2}{\beta_t^2}}\bar{\mathbf{z}}_t \qquad (17)$$

**Remark 4.3.** *Eq. 17 is refered as Markovian bridge in Zheng et al. (2024), and this setting can be used to reproduce the sampler in I2SB Liu et al. (2023), see Appendix C for more details.*

In our implementation, when we make $\epsilon_t = 0$ for the last two steps, Eq. (16) gets reduced to: $\mathbf{x}_{t-\Delta t} \approx \alpha_{t-\Delta t}\hat{\mathbf{x}}_0 + \beta_{t-\Delta t}\mathbf{x}_T + \gamma_{t-\Delta t}\hat{\mathbf{z}}_t$. For other steps, we apply Eq. (14) and let $\epsilon_t = \eta(\gamma_t\dot{\gamma}_t - \frac{\dot{\alpha}_t}{\alpha_t}\gamma_t^2)$, where $\eta$ is a constant. Putting all ingredients together leads to our sampler outlined in Algorithm 1.

---

**Algorithm 1** Denoising Diffusion Bridge Stochastic Sampler

---

**Require:** model $D_\theta(\mathbf{x}_t, \mathbf{x}_T, t)$, time steps $\{t_j\}_{j=0}^N$, input data distribution $\pi_{\text{cond}}$, scheduler $\alpha_t, \beta_t, \gamma_t, \epsilon_t, b$.
  1: Sample $\mathbf{x}_T \sim \pi_{\text{cond}}$, $\mathbf{n}_0 \sim \mathcal{N}(0, b^2\mathbf{I})$
  2: $\mathbf{x}_N = \mathbf{x}_T + \mathbf{n}_0$
  3: **for** $i = N, \ldots, 1$ **do**
  4:     $\hat{\mathbf{x}}_0 \leftarrow D_\theta(\mathbf{x}_i, \mathbf{x}_T, t_i)$
  5:     $\hat{\mathbf{z}}_i \leftarrow (\mathbf{x}_i - \alpha_{t_i}\hat{\mathbf{x}}_0 - \beta_{t_i}\mathbf{x}_N)/\gamma_{t_i}$
  6:     **if** $N \geq 2$ **then**
  7:         Sample $\bar{\mathbf{z}}_i \sim \mathcal{N}(0, \mathbf{I})$
  8:         $d_i \leftarrow \dot{\alpha}_{t_i}\hat{\mathbf{x}}_0 + \dot{\beta}_{t_i}\mathbf{x}_N + (\dot{\gamma}_{t_i} + \epsilon_{t_i}/\gamma_{t_i})\hat{\mathbf{z}}_i$
  9:         $\mathbf{x}_{i-1} \leftarrow \mathbf{x}_i + d_i(t_i - t_{i-1}) + \sqrt{2\epsilon_{t_i}(t_i - t_{i-1})}\bar{\mathbf{z}}_i$
10:     **else**
11:         $\mathbf{x}_{i-1} \leftarrow \alpha_{t_{i-1}}\hat{\mathbf{x}}_0 + \beta_{t_{i-1}}\hat{\mathbf{x}}_N + \gamma_{t_{i-1}}\hat{\mathbf{z}}_i$
12:     **end if**
13: **end for**

---

## 5 EXPERIMENTS

In this section, we demonstrate that SDBs achieve much better performance for I2I transition tasks, in terms of sample efficiency, image quality and conditional diversity. We evaluate on I2I translation tasks on Edges→Handbags (Isola et al., 2017) scaled to $64 \times 64$ pixels and DIODE-Outdoor scaled to $256 \times 256$ (Vasiljevic et al., 2019). For evaluation metrics, we use Fréchet Inception Distance (FID) (Heusel et al., 2017) for all experiments, and additionally measure Inception Scores (IS) (Barratt & Sharma, 2018), Learned Perceptual Image Patch Similarity (LPIPS) (Zhang et al., 2018), Mean Square Error (MSE), following previous works (Zheng et al., 2024; Linqi Zhou et al., 2023). In addition, we use AFD, Eq. 10, to measure conditional diversity, as further validated in Appendix A. Further details of the experiments and design guidelines are provided in Appendix E and D.

**Stochasticity control in sampling SDEs**. We evaluate different sampling algorithms in Fig. 3 (a), the results demonstrate that setting $\epsilon_t = 0$ and using Eq. (17) for the last 2 steps can significantly improve sampled image quality compared with simple Euler discretization and DDBM sampler. Furthermore, By specifically designing stochasticity control during sampling, our sampler surpasses the sampling results by DDBM and DBIM with the same pretrained model. The results are demonstrated in Table 2. We set the number of function evaluations (NFEs) from the set $[5, 10, 20]$ and select $\eta$ from the set $[0, 0.3, 0.5, 0.8, 1.0]$. We observed that our sampler achieves much lower FID compared to both DDBM sampler and DBIM sampler across all datasets and NFEs. Besides, the best performance achieved around $\eta = 0.3$, which is align with the total stochasticity added to the sampling process by original DDBM sampler (Linqi Zhou et al., 2023). The above results demonstrate the significance of designing the stochasicity added to the sampling process.

**Stochasticity control in transition kernel**. Despite the extensive design space available for the transition kernel, this paper focuses on Linear transition path with different strength of maximum

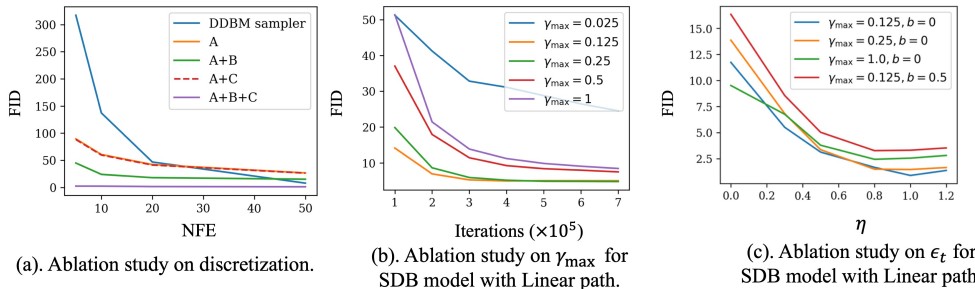

(a). Ablation study on discretization.

(b). Ablation study on $\gamma_{\max}$ for SDB model with Linear path.

(c). Ablation study on $\epsilon_t$ for SDB model with Linear path.

Figure 3: Ablation studies on discretization, $\gamma_{\max}$ and $\epsilon_t$. (a). We evaluate different discretization schemes on Edges2handbags ($64 \times 64$) dataset using DDBM-VP pretrained model, A represents simple Euler discretization in Eq. (14), B reprents setting $\epsilon_t = 0$ for the last 2 steps, C represents using Eq. (17) for $\epsilon_t = 0$. (b). Ablation study on $\gamma_{\max}$ evaluated by DIODE ($64 \times 64$) dataset. (c). Ablation study on $\epsilon_t$ through our SDB model with Linear path on Edges2handbags ($64 \times 64$) dataset, where $\epsilon_t = \eta(\gamma_t \dot{\gamma}_t - \frac{\dot{\alpha}_t}{\alpha_t} \gamma_t^2)$.

Table 2: Ablation Study of $\epsilon_t$ for DDBM-VP path via DDBM pretrained VP model (Evaluated by FID), where $\epsilon_t = \eta(\gamma_t \dot{\gamma}_t - \frac{\dot{\alpha}_t}{\alpha_t} \gamma_t^2)$.

| Sampler | $\eta$ | NFE | | | | | |
| | | 5 | 10 | 20 | 5 | 10 | 20 |
| | | Edges→Handbags ($64 \times 64$) | | | DIODE-Outdoor ($256 \times 256$) | | |
|---|---|---|---|---|---|---|---|
| DDBM (Linqi Zhou et al., 2023) | - | 317.22 | 137.15 | 46.74 | 328.33 | 151.93 | 41.03 |
| DBIM (Zheng et al., 2024) | - | 3.60 | 2.46 | 1.74 | 14.25 | 7.98 | 4.99 |
| SDB (Ours) | 0 | 10.89 | 11.45 | 11.69 | 77.31 | 84.68 | 87.34 |
| | 0.3 | **2.36** | **2.25** | **1.53** | **10.87** | **6.83** | **4.12** |
| | 0.5 | 10.21 | 7.17 | 4.18 | 18.94 | 12.91 | 8.07 |
| | 0.8 | 16.33 | 14.29 | 9.33 | 25.90 | 18.25 | 11.74 |
| | 1.0 | 18.78 | 17.61 | 13.59 | 30.62 | 21.64 | 14.08 |

stochasticity, i.e., $p_{t|0,T}(\mathbf{x}_t|\mathbf{x}_0, \mathbf{x}_T) = \mathcal{N}(\mathbf{x}_t; (1-t)\mathbf{x}_0 + t\mathbf{x}_T, \frac{1}{4}\gamma_{\max}^2 t(1-t)\mathbf{I})$. We conducted detailed ablation studies on $\gamma_{\max}$ and $\eta$ for the Linear path on DIODE ($64 \times 64$) dataset, as shown in Fig. 3 (b) and (c). The optimal values for $\gamma_{\max}$ were found to be $0.125$ and $0.25$, while the best performance for $\eta$ was achieved with $\eta = 0.8$ and $\eta = 1.0$. Performance deteriorates when either parameter is too small or too large. Based on the results of these ablation studies, we further trained SDB models on the Edges2handbags ($64 \times 64$) and DIODE ($256 \times 256$) datasets by taking $\gamma_{\max} \in \{0.125, 0.5\}$ and setting $\eta = 1.0$. The results are presented in Table 3. Our models establish a new benchmark for image quality, as evaluated by FID, IS and LPIPS. Despite our models having slightly higher MSEs compared to the baseline DDBM and DBIM, we believe that a larger MSE indicates that the generated images are distinct from their references, suggesting a richer diversity. We also provide the visualization of sampling process in Fig. 4.

**Stochasticity control in base distribution**. Through controlling stochasticity in the base distribution, we achieved a more diverse set of sample images, while this diversity comes at the cost of slightly higher FID scores and slower sampling speed. We show generated images in Fig. 5. More visualization can be found in Appendix F, which shows that by introducing booting noise to the input data distribution, the model can generate samples with more diverse colors and textures. Further quantitative results are presented in Table 4, confirming that our model surpasses the vanilla DDBM in terms of image quality, sample efficiency, and conditional diversity.

## 6 RELATED WORK

**Diffusion Bridge Models**. Diffusion bridges are faster diffusion processes that could learn the mapping between two random target distributions (Yifeng Shi et al., 2023; Stefano Peluchetti, 2023), demonstrating significant potential in various areas, such as protein docking (Somnath et al., 2023),

Table 3: Quantitative results in the I2I translation task edges2handbags ($64 \times 64$) and DIODE ($256 \times 256$) datasets. Our results were achieved by Linear transition kernel and setting $\eta = 1$.

| Model | NFE | Edges→handbags ($64 \times 64$) | | | | DIODE-Outdoor ($256 \times 256$) | | | |
|---|---|---|---|---|---|---|---|---|---|
| | | FID ↓ | IS ↑ | LPIPS ↓ | MSE | FID ↓ | IS ↑ | LPIPS ↓ | MSE |
| Pix2Pix (Isola et al., 2017) | 1 | 74.8 | 3.24 | 0.356 | 0.209 | 82.4 | 4.22 | 0.556 | 0.133 |
| DDIB (Su et al., 2022) | $\geq 40^{\dagger}$ | 186.84 | 2.04 | 0.869 | 1.05 | 242.3 | 4.22 | 0.798 | 0.794 |
| SDEdit (Meng et al., 2021) | $\geq 40$ | 26.5 | 3.58 | 0.271 | 0.510 | 31.14 | 5.70 | 0.714 | 0.534 |
| Rectified Flow (Liu et al., 2022b) | $\geq 40$ | 25.3 | 2.80 | 0.241 | 0.088 | 77.18 | 5.87 | 0.534 | 0.157 |
| I$^2$SB (Liu et al., 2023) | $\geq 40$ | 7.43 | 3.40 | 0.244 | 0.191 | 9.34 | 5.77 | 0.373 | 0.145 |
| DDBM (Linqi Zhou et al., 2023) | 118 | 1.83 | 3.73 | 0.142 | 0.040 | 4.43 | 6.21 | 0.244 | 0.084 |
| DBIM (Zheng et al., 2024) | 20 | 1.74 | 3.64 | 0.095 | 0.005 | 4.99 | 6.10 | 0.201 | 0.017 |
| | 5 | **0.89** | 4.10 | 0.049 | 0.024 | 12.97 | 5.49 | 0.269 | 0.074 |
| SDB ($\gamma_{\max} = 0.125$) | 10 | **0.67** | 4.11 | 0.045 | 0.024 | 10.12 | 5.56 | 0.255 | 0.076 |
| | 20 | **0.56** | 4.11 | 0.044 | 0.024 | 8.62 | 5.62 | 0.248 | 0.078 |
| | 5 | 1.46 | **4.21** | **0.040** | 0.016 | 4.16 | 5.83 | **0.104** | 0.029 |
| SDB ($\gamma_{\max} = 0.25$) | 10 | 1.38 | **4.22** | **0.038** | 0.017 | 3.44 | 5.86 | **0.098** | 0.029 |
| | 20 | 1.40 | **4.20** | **0.038** | 0.017 | 3.27 | 5.85 | **0.094** | 0.029 |

Figure 4: Visualization of the sampling process. The trajectories of $\hat{\mathbf{x}}_0$ suggest that in the initial stage of the diffusion model, more general features such as shape and color are constructed. As the process evolves, it progressively generates finer details and high-frequency elements like texture.

mean-field game (Liu et al., 2022a), I2I translation (Liu et al., 2023; Linqi Zhou et al., 2023). According to different design philosophies, DBMs can be divided into two groups: bridge matching and stochastic interpolants. The idea of bridge matching was first proposed by Peluchetti (2023), and can be viewed as a generalization of score matching (Song et al., 2020). Based on this, diffusion Schrödinger bridge matching (DSBM) has been developed for solving Schrödinger bridge problems Stefano Peluchetti (2023); Yifeng Shi et al. (2023). In addition, Liu et al. (2023) utilize bridge matching to perform image restoration tasks and noted benefits of stochasticity empirically, the experiments shows the new model is more efficient and interpretable than score-based generative models (Liu et al., 2023). Furthermore, our benchmark DDBM (Linqi Zhou et al., 2023) achieve significant improvement for various I2I translation tasks, DBIM (Zheng et al., 2024) improved the sampling algorithm for DDBM, significantly reducing sampling time while maintaining the same image quality. Flow Matching and Rectified Flow learn ODE models to facilitate transport between two empirically observed distributions (Lipman et al., 2022; Liu et al., 2022b). Stochastic interpolants further couple the base and target densities through SDEs (Albergo et al., 2023). Although our approach aligns with these methods, it diverges in various aspects. Unlike stochastic interpolation which models the data distribution $p_0$, our framework specifically targets sampling from the conditional distribution $p_{0|T}$, significantly simplifying both training and inference.

**Image-to-Image Translations**. Diffusion models have shown extraordinary performance in image synthesis. However, enhancing their capability in I2I translation presents several challenges, primarily the reduction of artifacts in translated images. To address this, DiffI2I mitigates misalignment and reduces artifacts in I2I translation tasks with fewer diffusion steps (Bin Xia et al., 2023). In the latent space, I2I translation is also achieved more quickly by S2ST (Or Greenberg et al., 2023), which consumes less memory. Various methods leverage different forms of guidance (Narek Tumanyan et al., 2023; Hyunsoo Lee et al., 2023), such as frequency control (Xiang Gao et al., 2024), to tackle these challenges. Another significant challenge is that I2I translation methods typically require joint training on both source and target domains, posing privacy concerns. Injecting-diffusion addresses this issue in unpaired I2I translation by extracting domain-independent content from the source image and fusing it into the target domain (Luying Li & Lizhuang Ma, 2023). To improve interpretability in unpaired translation, SDDM separates intermediate tangled generative distributions by decom-

Table 4: Quantitative results for sample efficiency, image quality, and conditional diversity. By adding stochasticity to the base distribution ($b > 0$), we achieved much better conditional diversity, evaluated by AFD. While the introduction of $b > 0$ results in a slight increase in FID and NFE, we believe this trade-off is advantageous in certain scenarios.

(a). DDBM-VP model + different samplers.

| Sampler | NFEs ↓ | FID ↓ | AFD ↑ |
|---------|--------|-------|-------|
| DDBM | 118 | 1.83 | 6.99 |
| DBIM | 5 | 3.60 | 5.63 |
| DBIM | 10 | 2.46 | 5.20 |
| DBIM | 20 | 1.74 | 5.84 |
| SDB | 5 | 2.36 | 5.11 |
| SDB | 10 | 2.25 | 5.70 |
| SDB | 20 | 1.53 | 6.04 |

(b). Our SDB models and samplers with different choices of $b$.

| $b$ | NFEs ↓ | FID ↓ | AFD ↑ |
|-----|--------|-------|-------|
| 0 | 5 | **0.89** | 6.00 |
| 0 | 10 | **0.67** | 6.05 |
| 0 | 20 | **0.56** | 6.25 |
| 0.5 | 5 | 3.31 | **8.53** |
| 0.5 | 10 | 2.07 | **9.35** |
| 0.5 | 20 | 1.74 | **9.65** |

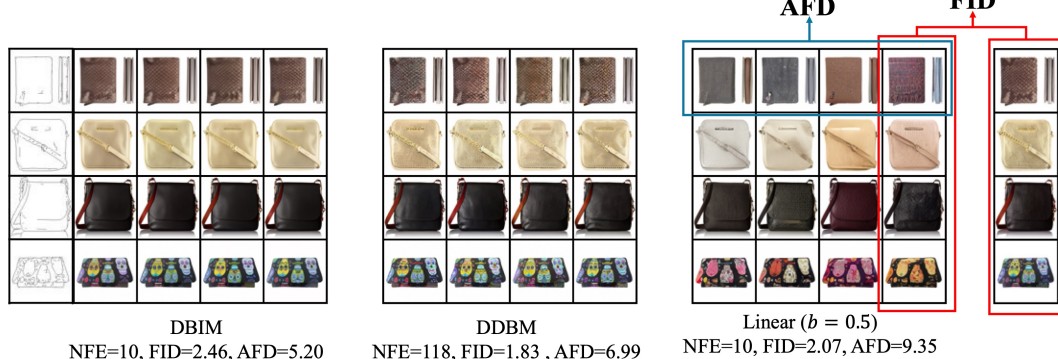

DBIM
NFE=10, FID=2.46, AFD=5.20

DDBM
NFE=118, FID=1.83 , AFD=6.99

Linear ($b = 0.5$)
NFE=10, FID=2.07, AFD=9.35

Figure 5: Visualization of conditional diversity via sampled images. While FID measures diversity within columns, AFD evaluates diversity across rows. The visualization further proved the effectiveness of AFD. More sampled images can be found in Appendix F.

posing the score function (Shurong Sun et al., 2023). Diffusion bridges are also popular due to their interpretability and ability to map between arbitrary distributions. DDIB employs an encoder trained on the source domain and a decoder trained on the target domain to establish Schrödinger Bridges (SBs) (Xu Su et al., 2022). Beomsu Kim et al. (2023) incorporates discriminators and regularization to learn an SB between unpaired data.

# 7 CONCLUSION

In this study, we introduced the Stochasticity-controlled Diffusion Bridge (SDB), a framework designed to facilitate translation between two arbitrary distributions. By strategically managing stochasticity in the base distribution, transition kernel, and sampling SDEs, our approach improves image quality, sampling efficiency, and conditional diversity, allowing for the tailored design of diffusion bridge models across a range of tasks. This work is the first to derive sampling SDEs of $q(\mathbf{X}_t \mid \mathbf{x}_T)$ for arbitrary Gaussian transition kernels of the form $\mathcal{N}(x_t; \alpha_t \mathbf{x}_0 + \beta_t \mathbf{x}_T, \gamma_t^2 \mathbf{I})$. Additionally, our approach is the first to highlight the issue of lacking conditional diversity in diffusion bridge models and to resolve it by introducing stochasticity into the base distribution. We highlighted the importance of stochasticity control (SC) and addressed challenges associated with singularity through score reparameterization and specially designed discretization. Our results demonstrate that a simple linear bridge configuration can set new benchmarks in image quality, sampling efficiency and conditional diversity, as evidenced by our experiments with $64 \times 64$ edges2handbags and $256 \times 256$ DIODE-outdoor I2I translation tasks. Despite these advancements, we acknowledge that the optimal stochasticity may vary from one scenario to another, indicating a rich avenue for further exploration and refinement in future work.

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

Table 5: Evaluation for generative models: ImageNet-1-mode, ImageNet-2-modes, ImageNet-5-modes, and ImageNet-10-modes.

| Model | ImageNet-1-mode | ImageNet-2-modes | ImageNet-5-modes | ImageNet-10-modes |
|---|---|---|---|---|
| FID | 58.30 | 57.34 | 57.78 | 57.26 |
| AFD | 0 | 8.14 | 12.84 | 14.47 |

# A  AFD VALIDATION

In this section, we thoroughly validate the effectiveness of our proposed metric, AFD, for measuring conditional diversity and demonstrate its role as a complementary metric to FID. In unconditional generation scenarios, the FID is widely used to evaluate the diversity of generated images. While low FID scores generally indicate high diversity across the entire dataset, they do not necessarily imply high conditional diversity. For instance, we observed that samples generated by the DDBM model often lack diversity when conditioned on edge images, despite achieving very low FID scores. To address this limitation, we introduce the concept of conditional diversity and propose a corresponding metric to quantify it.

The first question is why FID failed to measure the conditional diversity. To illustrate the limitations of FID in capturing conditional diversity, consider an extreme case: if the images generated by a generative model are identical to a set of baseline images, the FID score can be very low since the two distributions are indistinguishable. However, this scenario does not reflect diversity within the conditional outputs.

To further support our point, we designed two classes of pseudo-generative models capable of controlling the diversity of the generated images, which are further validated by FID and AFD. The experiments are evaluated on Imagenet dataset (Deng et al., 2009).

## A.1  PSEUDO-GENERATIVE MODELS BY RANDOM SELECTION

We designed four pseudo-generative models: ImageNet-1-mode, ImageNet-2-modes, ImageNet-5-modes, and ImageNet-10-modes. The experimental setup is as follows:

- We selected 11,000 samples from the ImageNet validation dataset, randomly choosing 11 images per class.
- From these, we designated 1,000 images as the "real" set, while the remaining images served as the source pool for the generative models.
- Each ImageNet-k-modes model simulates a generative process by randomly sampling images from a pool of $k$ distinct images within a given class.

We present sampled images in Fig. 6, where it is evident that the ImageNet-10-modes model generates images with the highest conditional diversity. To quantify this, we conducted experiments to calculate both FID and AFD for the four generative models. The results are summarized in Table 5. While the FID scores are nearly identical across all models, the AFD values increase as the conditional diversity of the generative models improves. This highlights that AFD is a more effective metric for capturing conditional diversity than FID.

## A.2  PSEUDO-GENERATIVE MODELS BY STRONG AUGMENTATION

Strong augmentation has been widely used in computer vision to generate synthetic data while preserving its underlying semantics (Chen et al., 2020; Zbontar et al., 2021; Sohn et al., 2020; Berthelot et al., 2019). The intensity of augmentation can be adjusted, with higher intensities producing more diverse images. To further validate our proposed metric, AFD, as a measure of diversity, we construct pseudo-generative models using strong augmentation.

We selected 1,000 images from the ImageNet-1k dataset, one from each category. These images were subjected to data augmentation, specifically using ColorJitter, with varying magnitudes to enhance diversity. For each image, the augmentation was applied 16 times, creating an augmented

ImageNet-1-mode: FID=58.30, AFD=0

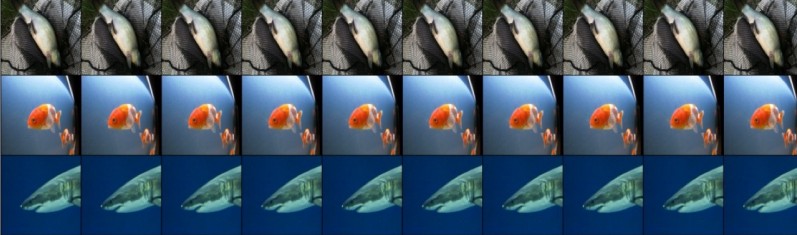

ImageNet-2-modes: FID=57.34, AFD=8.14

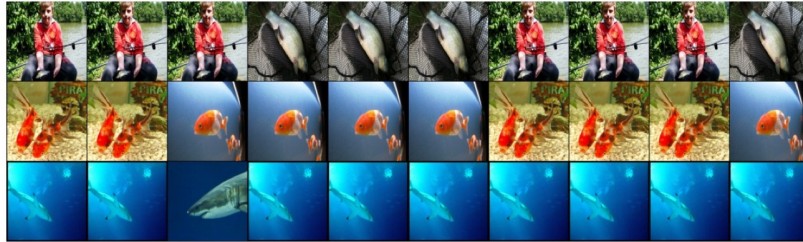

ImageNet-5-modes: FID=57.78, AFD=12.84

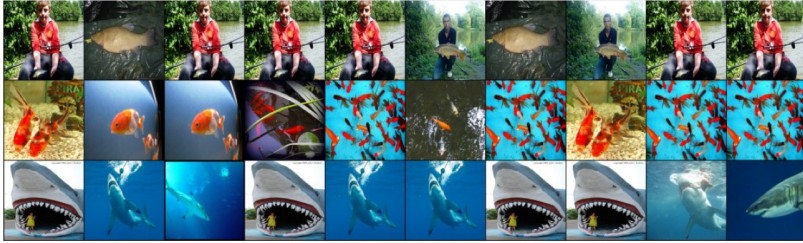

ImageNet-10-modes: FID=57.26, AFD=14.47

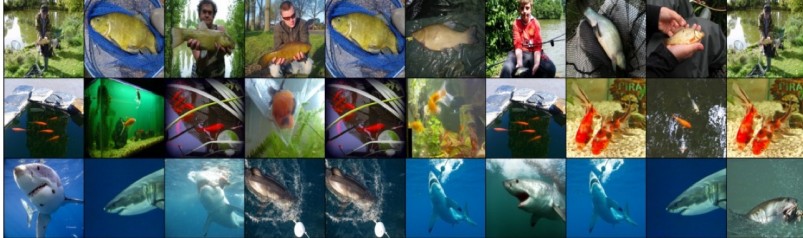

Figure 6: Sampled images from 4 generative models: ImageNet-1-mode, ImageNet-2-modes, ImageNet-5-modes, ImageNet-10-modes.

Table 6: AFD results across different augmentation magnitudes

| Augmentation magnitude | 0.1 | 0.2 | 0.3 | 0.4 | 0.5 | 0.6 | 0.7 | 0.8 |
|---|---|---|---|---|---|---|---|---|
| AFD | 2.16 | 3.77 | 5.13 | 6.16 | 6.98 | 7.63 | 8.22 | 9.01 |
| FID | 0.20 | 2.95 | 7.02 | 11.62 | 16.33 | 20.84 | 25.12 | 28.89 |

dataset for each magnitude setting. We then calculated the AFD for these augmented datasets to evaluate the relationship between dataset diversity (as influenced by augmentation magnitude) and the AFD value.

Table 6 summarizes the AFD results across various augmentation magnitude settings. The results show that as diversity increases, AFD values also rise, further confirming that the proposed AFD metric is a reliable indicator of image diversity.

# B  PROOFS

There are infinitely many pinned processes characterized by the Gaussian transition kernel $p_{t|0,T}(\mathbf{x}_t \mid \mathbf{x}_0, \mathbf{x}_T) = \mathcal{N}(\mathbf{x}_t; \alpha_t \mathbf{x}_0 + \beta_t \mathbf{x}_T, \gamma_t^2 \mathbf{I})$. Specifically, we formalize the pinned process as a linear Itô SDE, as presented in Lemma 3.

**Lemma 3.** *There exist a linear Itô SDE*

$$d\mathbf{X}_t = [f_t \mathbf{X}_t + s_t \mathbf{x}_T]dt + g_t d\mathbf{W}_t, \quad \mathbf{X}_0 = \mathbf{x}_0, \tag{18}$$

*where $f_t = \frac{\dot{\alpha}_t}{\alpha_t}, \quad s_t = \dot{\beta}_t - \frac{\dot{\alpha}_t}{\alpha_t}\beta_t, \quad g_t = \sqrt{2(\gamma_t \dot{\gamma}_t - \frac{\dot{\alpha}_t}{\alpha_t}\gamma_t^2)}$, that has a Gaussian marginal distribution $\mathcal{N}\left(\mathbf{x}_t; \alpha_t \mathbf{x}_0 + \beta_t \mathbf{x}_T, \gamma_t^2 \mathbf{I}\right)$.*

Given the pinned process (18), we can sample from the conditional distribution $p_{0|T}(\mathbf{x}_0|\mathbf{x}_T)$ by solving the reverse SDE or ODE from $t = T$ to $t = 0$:

$$d\mathbf{X}_t = \left[f_t \mathbf{X}_t + s_t \mathbf{x}_T - g_t^2 \nabla_{\mathbf{X}_t} \log p_t(\mathbf{X}_t|\mathbf{x}_T)\right] dt + g_t d\mathbf{W}_t, \quad \mathbf{X}_T = \mathbf{x}_T, \tag{19}$$

$$d\mathbf{X}_t = \left[f_t \mathbf{X}_t + s_t \mathbf{x}_T - \frac{1}{2}g_t^2 \nabla_{\mathbf{X}_t} \log p_t(\mathbf{X}_t|\mathbf{x}_T)\right] dt \quad \mathbf{X}_T = \mathbf{x}_T, \tag{20}$$

where the score $\nabla_{\mathbf{X}_t} \log p_t(\mathbf{X}_t|\mathbf{x}_T)$ can be estimated by score matching objective (8). To improve training stability, we introduced score reparameterization in Sec. 4.1.

**Lemma 1.** *There exist a linear Itô SDE*

$$d\mathbf{X}_t = [f_t \mathbf{X}_t + s_t \mathbf{x}_T]dt + g_t d\mathbf{W}_t, \quad \mathbf{X}_0 = \mathbf{x}_0, \tag{21}$$

*where $f_t = \frac{\dot{\alpha}_t}{\alpha_t}, \quad s_t = \dot{\beta}_t - \frac{\dot{\alpha}_t}{\alpha_t}\beta_t, \quad g_t = \sqrt{2(\gamma_t \dot{\gamma}_t - \frac{\dot{\alpha}_t}{\alpha_t}\gamma_t^2)}$, that has a Gaussian marginal distribution $\mathcal{N}\left(\mathbf{x}_t; \alpha_t \mathbf{x}_0 + \beta_t \mathbf{x}_T, \gamma_t^2 \mathbf{I}\right)$.*

*Proof.* Let $\mathbf{m}_t$ denote the mean function of the given Itô SDE, then we have $\frac{d\mathbf{m}_t}{dt} = f_t \mathbf{m}_t + s_t \mathbf{x}_T$. Given the transition kernel, the mean function $\mathbf{m}_t = \alpha_t \mathbf{x}_0 + \beta_t \mathbf{x}_T$, therefore,

$$\dot{\alpha}_t \mathbf{x}_0 + \dot{\beta}_t \mathbf{x}_T = f_t(\alpha_t \mathbf{x}_0 + \beta_t \mathbf{x}_T) + s_t \mathbf{x}_T. \tag{22}$$

Matching the above equation:

$$f_t = \frac{\dot{\alpha}_t}{\alpha_t}, s_t = \dot{\beta}_t - \beta_t \frac{\dot{\alpha}_t}{\alpha_t}. \tag{23}$$

Further, For the variance $\gamma_t^2$ of the process, the dynamics are given by:

$$\frac{d\gamma_t^2}{dt} = 2f_t \gamma_t^2 + g_t^2. \tag{24}$$

Solving for $g_t^2$, we substitute $f_t = \frac{\dot{\alpha}_t}{\alpha_t}$:

$$g_t^2 = \frac{d\gamma_t^2}{dt} - 2\frac{\dot{\alpha}_t}{\alpha_t}\gamma_t^2 \tag{25}$$

Therefore,

$$g_t = \sqrt{2(\gamma_t \dot{\gamma}_t - \frac{\dot{\alpha}_t}{\alpha_t}\gamma_t^2)}. \tag{26}$$

□

For dynamics described by ODE $d\mathbf{X}_t = \mathbf{u}_t dt$, we can identify the entire class of SDEs that maintain the same marginal distributions, as detailed in Lemma 2. This enables us to control the stochasticity during sampling by appropriately designing $\epsilon_t$.

**Lemma 2.** *Consider a continuous dynamics given by ODE of the form: $d\mathbf{X}_t = \mathbf{u}_t dt$, with the density evolution $p_t(\mathbf{X}_t)$. Then there exists forward SDEs and backward SDEs that match the marginal distribution $p_t$. The forward SDEs are given by: $d\mathbf{X}_t = (\mathbf{u}_t + \epsilon_t \nabla \log p_t) dt + \sqrt{2\epsilon_t} d\mathbf{W}_t, \epsilon_t > 0$. The backward SDEs are given by: $d\mathbf{X}_t = (\mathbf{u}_t - \epsilon_t \nabla \log p_t) dt + \sqrt{2\epsilon_t} d\mathbf{W}_t, \epsilon_t > 0$.*

*Proof.* For the forward SDEs, the Fokker-Planck equations are given by:

$$\frac{\partial p_t(\mathbf{X}_t)}{\partial t} = -\nabla \cdot \left[ (\mathbf{u}_t + \epsilon_t \nabla \log p_t) \, p_t(\mathbf{X}_t) \right] + \epsilon_t \nabla^2 p_t(\mathbf{X}_t) \tag{27}$$

$$= -\nabla \cdot [\mathbf{u}_t p_t(\mathbf{X}_t)] - \nabla \cdot [\epsilon_t (\nabla \log p_t) p_t(\mathbf{X}_t)] + \epsilon_t \nabla^2 p_t(\mathbf{X}_t) \tag{28}$$

$$= -\nabla \cdot [\mathbf{u}_t p_t(\mathbf{X}_t)] - \epsilon_t \nabla \cdot [\nabla p_t(\mathbf{X}_t)] + \epsilon_t \nabla^2 p_t(\mathbf{X}_t) \tag{29}$$

$$= -\nabla \cdot [\mathbf{u}_t p_t(\mathbf{X}_t)] \,. \tag{30}$$

This is exactly the Fokker-Planck equation for the original deterministic ODE $d\mathbf{X}_t = \mathbf{u}_t \, dt$. Therefore, the forward SDE maintains the same marginal distribution $p_t(\mathbf{X}_t)$ as the original ODE.

Now consider the backward SDEs, the Fokker-Planck equations become:

$$\frac{\partial p_t(\mathbf{X}_t)}{\partial t} = -\nabla \cdot \left[ (\mathbf{u}_t - \epsilon_t \nabla \log p_t) \, p_t(\mathbf{X}_t) \right] - \epsilon_t \nabla^2 p_t(\mathbf{X}_t) \tag{31}$$

$$= -\nabla \cdot [\mathbf{u}_t p_t(\mathbf{X}_t)] + \nabla \cdot [\epsilon_t (\nabla \log p_t) p_t(\mathbf{X}_t)] - \epsilon_t \nabla^2 p_t(\mathbf{X}_t) \tag{32}$$

$$= -\nabla \cdot [\mathbf{u}_t p_t(\mathbf{X}_t)] \,. \tag{33}$$

This is again the Fokker-Planck equation corresponding to the original deterministic ODE $d\mathbf{X}_t = \mathbf{u}_t \, dt$. Therefore, the backward SDE also maintains the same marginal distribution $p_t(\mathbf{X}_t)$.

$\square$

**Theorem 3.** *Suppose the transition kernel of a diffusion process is given by $p_{t|0,T}(\mathbf{x}_t \mid \mathbf{x}_0, \mathbf{x}_T) = \mathcal{N}(\mathbf{x}_t; \alpha_t \mathbf{x}_0 + \beta_t \mathbf{x}_T, \gamma_t^2 \mathbf{I})$, then the evolution of conditional probability $q(\mathbf{X}_t|\mathbf{x}_T)$ has a class of time reverse sampling SDEs of the form:*

$$d\mathbf{X}_t = \left[ \dot{\alpha}_t \hat{\mathbf{x}}_0 + \dot{\beta}_t \mathbf{x}_T - (\dot{\gamma}_t \gamma_t + \epsilon_t) \nabla_{\mathbf{X}_t} \log p_t(\mathbf{X}_t|\mathbf{x}_T) \right] dt + \sqrt{2\epsilon_t} d\mathbf{W}_t \quad \mathbf{X}_T = \mathbf{x}_T. \tag{34}$$

*Proof.* Recall Eqs. (19) 20 and Lemma 2,

$$d\mathbf{X}_t = \left[ \frac{\dot{\alpha}_t}{\alpha_t} \mathbf{x}_t + (\dot{\beta}_t - \frac{\dot{\alpha}_t}{\alpha_t} \beta_t) \mathbf{x}_T - (\gamma_t \dot{\gamma}_t - \frac{\dot{\alpha}_t}{\alpha_t} \gamma_t^2 + \epsilon_t) \nabla_{\mathbf{x}_t} \log p_t(\mathbf{x}_t|\mathbf{x}_T) \right] dt + \sqrt{2\epsilon_t} d\mathbf{w}_t. \tag{35}$$

$\square$

Next we take the reparameterized score 12 into 35:

$$d\mathbf{X}_t = \left[ \frac{\dot{\alpha}_t}{\alpha_t} \mathbf{X}_t + (\dot{\beta}_t - \frac{\dot{\alpha}_t}{\alpha_t} \beta_t) \mathbf{x}_T - (\gamma_t \dot{\gamma}_t - \frac{\dot{\alpha}_t}{\alpha_t} \gamma_t^2 + \epsilon_t) \frac{\alpha_t \hat{\mathbf{x}}_0 + \beta_t \mathbf{x}_T - \mathbf{X}_t}{\gamma_t^2} \right] dt + \sqrt{2\epsilon_t} d\mathbf{w}_t \tag{36}$$

$$= \left[ \dot{\alpha}_t \hat{\mathbf{x}}_0 + \dot{\beta}_t \mathbf{x}_T - (\gamma_t \dot{\gamma}_t + \epsilon_t) \frac{\alpha_t \hat{\mathbf{x}}_0 + \beta_t \mathbf{x}_T - \mathbf{X}_t}{\gamma_t^2} \right] dt + \sqrt{2\epsilon_t} d\mathbf{w}_t \tag{37}$$

$$= \left[ \dot{\alpha}_t \hat{\mathbf{x}}_0 + \dot{\beta}_t \mathbf{x}_T - (\dot{\gamma}_t + \frac{\epsilon_t}{\gamma_t}) \frac{\alpha_t \hat{\mathbf{x}}_0 + \beta_t \mathbf{x}_T - \mathbf{X}_t}{\gamma_t} \right] dt + \sqrt{2\epsilon_t} d\mathbf{w}_t \tag{38}$$

$$= \left[ \dot{\alpha}_t \hat{\mathbf{x}}_0 + \dot{\beta}_t \mathbf{x}_T - (\dot{\gamma}_t + \frac{\epsilon_t}{\gamma_t}) \hat{\mathbf{z}} \right] dt + \sqrt{2\epsilon_t} d\mathbf{w}_t. \tag{39}$$

**Theorem 4.** *Let* $(\mathbf{x}_0, \mathbf{x}_T) \sim \pi_0(\mathbf{x}_0, \mathbf{x}_T)$, $\mathbf{x}_t \sim p_t(\mathbf{x}|\mathbf{x}_0, \mathbf{x}_T)$, *Given the transition kernel:* $p(\mathbf{x}_t \mid \mathbf{x}_0, \mathbf{x}_T) = \mathcal{N}\left(\mathbf{x}_t; \alpha_t \mathbf{x}_0 + \beta_t \mathbf{x}_T, \gamma_t^2 \mathbf{I}\right)$, *if* $\hat{\mathbf{x}}_0(\mathbf{x}_t, \mathbf{x}_T, t)$ *is a denoiser function that minimizes the expected* $L_2$ *denoising error for samples drawn from* $\pi_0(\mathbf{x}_0, \mathbf{x}_T)$:

$$\hat{\mathbf{x}}_0(\mathbf{x}_t, \mathbf{x}_T, t) = \arg\min_{D(\mathbf{x}_t, \mathbf{x}_T, t)} \mathbb{E}_{\mathbf{x}_0, \mathbf{x}_T, \mathbf{x}_t} \left[\lambda(t)\|D(\mathbf{x}_t, \mathbf{x}_T, t) - \mathbf{x}_0\|_2^2\right], \tag{40}$$

*then the score has the following relationship with* $\hat{\mathbf{x}}_0(\mathbf{x}_t, \mathbf{x}_T, t)$:

$$\nabla_{\mathbf{x}_t} \log p_t(\mathbf{x}_t|\mathbf{x}_T) = \frac{\alpha_t \hat{\mathbf{x}}_0(\mathbf{x}_t, \mathbf{x}_T, t) + \beta_t \mathbf{x}_T - \mathbf{x}_t}{\gamma_t^2}. \tag{41}$$

*Proof.*

$$\mathcal{L}(D) = \mathbb{E}_{(\mathbf{x}_0, \mathbf{x}_T) \sim \pi_0(\mathbf{x}_0, \mathbf{x}_T)} \mathbb{E}_{\mathbf{x}_t \sim p_t(\mathbf{x}_t|\mathbf{x}_0, \mathbf{x}_T)} \|D(\mathbf{x}_t) - \mathbf{x}_0\|_2^2 \tag{42}$$

$$= \int_{\mathbb{R}^d} \int_{\mathbb{R}^d} \underbrace{\int_{\mathbb{R}^d} p_t(\mathbf{x}_t|\mathbf{x}_0, \mathbf{x}_T) \pi_0(\mathbf{x}_0, \mathbf{x}_T) \|D(\mathbf{x}_t) - \mathbf{x}_0\|_2^2 \, \mathrm{d}\mathbf{x}_0}_{=:\mathcal{L}(D; \mathbf{x}_t, \mathbf{x}_T)} \, \mathrm{d}\mathbf{x}_T \mathrm{d}\mathbf{x}_t, \tag{43}$$

$$\mathcal{L}(D; \mathbf{x}_t, \mathbf{x}_T) = \int_{\mathbb{R}^d} p_t(\mathbf{x}_t|\mathbf{x}_0, \mathbf{x}_T) \pi_0(\mathbf{x}_0, \mathbf{x}_T) \|D(\mathbf{x}_t) - \mathbf{x}_0\|_2^2 \, \mathrm{d}\mathbf{x}_0, \tag{44}$$

we can minimize $\mathcal{L}(D)$ by minimizing $\mathcal{L}(D; \mathbf{x}_t, \mathbf{x}_T)$ independently for each $\{\mathbf{x}_t, \mathbf{x}_T\}$ pair.

$$D^*(\mathbf{x}_t, \mathbf{x}_T) = \arg\min_{D(\mathbf{x}_t)} \mathcal{L}(D; \mathbf{x}_t, \mathbf{x}_T) \tag{45}$$

$$\mathbf{0} = \nabla_{D(\mathbf{x}_t, \mathbf{x}_T)} [\mathcal{L}(D; \mathbf{x}_t, \mathbf{x}_T)] \tag{46}$$

$$= \int_{\mathbb{R}^d} p_t(\mathbf{x}_t|\mathbf{x}_0, \mathbf{x}_T) \pi_0(\mathbf{x}_0, \mathbf{x}_T) 2[D(\mathbf{x}, \mathbf{x}_T) - \mathbf{x}_0] \, \mathrm{d}\mathbf{x}_0 \tag{47}$$

$$= 2[D(\mathbf{x}_t, \mathbf{x}_T) \int_{\mathbb{R}^d} p_t(\mathbf{x}_t|\mathbf{x}_0, \mathbf{x}_T) \pi_0(\mathbf{x}_0, \mathbf{x}_T) \, \mathrm{d}\mathbf{x}_0 - \int_{\mathbb{R}^d} p_t(\mathbf{x}_t|\mathbf{x}_0, \mathbf{x}_T) \pi_0(\mathbf{x}_0, \mathbf{x}_T) \mathbf{x}_0 \, \mathrm{d}\mathbf{x}_0] \tag{48}$$

$$= 2[D(\mathbf{x}) p_t(\mathbf{x}_t, \mathbf{x}_T) - \int_{\mathbb{R}^d} p_t(\mathbf{x}_t|\mathbf{x}_0, \mathbf{x}_T) \pi_0(\mathbf{x}_0, \mathbf{x}_T) \mathbf{x}_0 \, \mathrm{d}\mathbf{x}_0], \tag{49}$$

$$D^*(\mathbf{x}_t, \mathbf{x}_T) = \int_{\mathbb{R}^d} \frac{p_t(\mathbf{x}_t|\mathbf{x}_0, \mathbf{x}_T) \pi_0(\mathbf{x}_0, \mathbf{x}_T) \mathbf{x}_0}{p_t(\mathbf{x}_t, \mathbf{x}_T)} \, \mathrm{d}\mathbf{x}_0, \tag{50}$$

$$\nabla_{\mathbf{x}_t} \log p_t(\mathbf{x}_t|\mathbf{x}_T) = \frac{\nabla_{\mathbf{x}_t} p_t(\mathbf{x}_t, \mathbf{x}_T)}{p_t(\mathbf{x}_t, \mathbf{x}_T)} \tag{51}$$

$$= \frac{\int \nabla_{\mathbf{x}_t} p_t(\mathbf{x}_t|\mathbf{x}_T, \mathbf{x}_0) \pi_0(\mathbf{x}_0, \mathbf{x}_T) d\mathbf{x}_0}{p_t(\mathbf{x}_t, \mathbf{x}_T)} \tag{52}$$

$$= -\int \frac{\mathbf{x}_t - \alpha_t \mathbf{x}_0 - \beta_t \mathbf{x}_T}{\gamma^2} \frac{p_t(\mathbf{x}_t|\mathbf{x}_0, \mathbf{x}_T) \pi_0(\mathbf{x}_0, \mathbf{x}_T)}{p_t(\mathbf{x}_t, \mathbf{x}_T)} d\mathbf{x}_0 \tag{53}$$

$$= \frac{\alpha_t D^*(\mathbf{x}_t, \mathbf{x}_T) + \beta_t \mathbf{x}_T - \mathbf{x}_t}{\gamma^2}. \tag{54}$$

Thus we conclude the proof.

$\square$

## C REFRAMING PREVIOUS METHODS IN OUR FRAMEWORK

We draw a link between our framework and the diffusion bridge models used in DDBM.

### C.1 DDBM-VE

DDBM-VE can be reformulated in our framework as we set :

$$\alpha_t = s_t(1 - \frac{\sigma_t^2}{\sigma_T^2}), \beta_t = \frac{s_t\sigma_t^2}{s_1\sigma_T^2}, \gamma_t = \sigma_t s_t\sqrt{(1 - \frac{\sigma_t^2}{\sigma_T^2})} \tag{55}$$

*Proof.* In the origin DDBM paper, the evolution of conditional probability $q(\mathbf{x}_t|\mathbf{x}_T)$ has a time reversed SDE of the form:

$$d\mathbf{X}_t = \left[\bar{\mathbf{f}}_t(\mathbf{X}_t) - \bar{g}_t^2\bar{\mathbf{h}}_t(\mathbf{X}_t) - \bar{g}_t^2\mathbf{s}_t(\mathbf{X}_t)\right]dt + \bar{g}_t d\hat{\mathbf{W}}_t, \tag{56}$$

and an associated probability flow ODE

$$d\mathbf{X}_t = \left[\bar{\mathbf{f}}_t(\mathbf{X}_t) - \bar{g}_t^2\bar{\mathbf{h}}_t(\mathbf{X}_t) - \frac{1}{2}\bar{g}_t^2\mathbf{s}_t(\mathbf{X}_t)\right]dt. \tag{57}$$

Compare Eqs. (56) and 57 with Lemma 3. We only need to prove:

$$\bar{\mathbf{f}}_t(\mathbf{X}_t) - \bar{g}_t^2\bar{\mathbf{h}}_t(\mathbf{X}_t) = f_t\mathbf{X}_t + s_t\mathbf{x}_T, \bar{g}_t = g_t. \tag{58}$$

In the original paper,

$$\bar{\mathbf{f}}_t(\mathbf{X}_t) = 0, \bar{g}_t^2 = \frac{d}{dt}\sigma_t^2, \bar{\mathbf{h}}_t(\mathbf{X}_t) = \frac{\mathbf{x}_T - \mathbf{x}_t}{\sigma_T^2 - \sigma_t^2}. \tag{59}$$

Therefore,

$$\bar{\mathbf{f}}_t(\mathbf{X}_t) - \bar{g}_t^2\bar{\mathbf{h}}_t(\mathbf{X}_t) = \frac{2\sigma_t\dot{\sigma}_t(\mathbf{x}_T - \mathbf{x}_t)}{\sigma_T^2 - \sigma_t^2}, \bar{g}_t^2 = 2\dot{\sigma}_t\sigma_t. \tag{60}$$

In our framework, $f_t, s_t, g_t^2$ can be calculated:

$$f_t = \frac{\dot{\alpha}_t}{\alpha_t} = \frac{d}{dt}\log\alpha_t = \frac{d}{dt}\log\frac{\sigma_T^2 - \sigma_t^2}{\sigma_T^2} = \frac{-2\sigma_t\dot{\sigma}_t}{\sigma_T^2 - \sigma_t^2}, \tag{61}$$

$$s_t = \dot{\beta}_t - \frac{\dot{\alpha}_t}{\alpha_t}\beta_t = \frac{2\sigma_t\dot{\sigma}_t}{\sigma_T^2} + \frac{2\sigma_t\dot{\sigma}_t}{\sigma_T^2 - \sigma_t^2}\cdot\frac{\sigma_t^2}{\sigma_T^2} = \frac{2\sigma_t\dot{\sigma}_t}{\sigma_T^2 - \sigma_t^2}. \tag{62}$$

$$g_t^2 = 2(\gamma_t\dot{\gamma}_t - \frac{\dot{\alpha}_t}{\alpha_t}\gamma_t^2) = 2\gamma_t^2\left(\frac{\dot{\gamma}_t}{\gamma_t} - \frac{\dot{\alpha}_t}{\alpha_t}\right) = \gamma_t^2\left(\frac{(\sigma_T^2 - 2\sigma_t^2)\dot{\sigma}_t}{(\sigma_T^2 - \sigma_t^2)\sigma_t} + \frac{2\dot{\sigma}_t\sigma_t}{\sigma_T^2 - \sigma_t^2}\right) = 2\sigma_t\dot{\sigma}_t. \tag{63}$$

Therefore,

$$f_t\mathbf{X}_t + s_t\mathbf{x}_T = \frac{2\sigma_t\dot{\sigma}_t(\mathbf{x}_T - \mathbf{x}_t)}{\sigma_T^2 - \sigma_t^2} = \bar{\mathbf{f}}_t(\mathbf{X}_t) - \bar{g}_t^2\bar{\mathbf{h}}_t(\mathbf{X}_t), \quad \bar{g}_t = g_t, \tag{64}$$

which matches the formulation in DDBM.

$\square$

## C.2   DDBM-VP

DDBM-VP can be reformulated in our framework as we set :

$$\alpha_t = a_t(1 - \frac{\sigma_t^2 a_1^2}{\sigma_1^2 a_t^2}), \beta_t = \frac{\sigma_t^2 a_1}{\sigma_1^2 a_t}, \gamma_t = \sqrt{\sigma_t^2(1 - \frac{\sigma_t^2 a_1^2}{\sigma_1^2 a_t^2})}. \tag{65}$$

*Proof.* In the original DDBM-VP setting,

$$\bar{\mathbf{f}}_t(\mathbf{X}_t) = \frac{d \log a_t}{dt}\mathbf{x}_t, \tag{66}$$

$$\bar{g}_t^2 = 2\sigma_t\dot{\sigma}_t - 2\frac{\dot{a}_t}{a_t}\sigma_t^2 = \frac{2\sigma_t\dot{\sigma}_t a_t - 2\sigma_t^2\dot{a}_t}{a_t}, \tag{67}$$

$$\bar{\mathbf{h}}_t(\mathbf{X}_t) = \frac{(a_t/a_1)\mathbf{x}_T - \mathbf{x}_t}{\sigma_t^2(\text{SNR}_t/\text{SNR}_1 - 1)} = \frac{a_1 a_t \mathbf{x}_T - a_1^2 \mathbf{x}_t}{\sigma_1^2 a_t^2 - \sigma_t^2 a_1^2}. \tag{68}$$

Therefore,

$$\bar{\mathbf{f}}_t(\mathbf{X}_t) - \bar{g}_t^2\bar{\mathbf{h}}_t(\mathbf{X}_t) = \left[\frac{\dot{a}_t}{a_t} - \frac{2\sigma_t a_1^2(\dot{\sigma}_t a_t - \sigma_t\dot{a}_t)}{a_t(\sigma_1^2 a_t^2 - \sigma_t^2 a_1^2)}\right]\mathbf{x}_t + \frac{2\sigma_t a_1(\dot{\sigma}_t a_t - \sigma_t\dot{a}_t)}{\sigma_1^2 a_t^2 - \sigma_t^2 a_1^2}\mathbf{x}_T. \tag{69}$$

In our framework, $f_t, s_t, g_t^2$ can be calculated:

$$f_t = \frac{\dot{\alpha}_t}{\alpha_t} = \frac{d}{dt}\log \alpha_t \tag{70}$$

$$= \frac{d}{dt}\log \frac{\sigma_1^2 a_t^2 - \sigma_t^2 a_1^2}{\sigma_1^2 a_t} \tag{71}$$

$$= \frac{2\sigma_1^2 a_t\dot{a}_t - 2a_1^2\sigma_t\dot{\sigma}_t}{\sigma_1^2 a_t^2 - \sigma_t^2 a_1^2} - \frac{\dot{a}_t}{a_t} \tag{72}$$

$$= \frac{\dot{a}_t}{a_t} - \frac{2a_1^2\sigma_t(a_t\dot{\sigma}_t - \dot{a}_t\sigma_t)}{a_t(\sigma_1^2 a_t^2 - \sigma_t^2 a_1^2)}, \tag{73}$$

$$s_t = \dot{\beta}_t - \frac{\dot{\alpha}_t}{\alpha_t}\beta_t = \beta_t(\frac{\dot{\beta}_t}{\beta_t} - \frac{\dot{\alpha}_t}{\alpha_t}) \tag{74}$$

$$= \frac{\sigma_t^2 a_1}{\sigma_1^2 a_t}\left(\frac{2\dot{\sigma}_t}{\sigma_t} - \frac{2\sigma_1^2 a_t\dot{a}_t - 2a_1^2\sigma_t\dot{\sigma}_t}{\sigma_1^2 a_t^2 - \sigma_t^2 a_1^2}\right) \tag{75}$$

$$= \frac{2\sigma_t a_1(\dot{\sigma}_t a_t - \sigma_t\dot{a}_t)}{\sigma_1^2 a_t^2 - \sigma_t^2 a_1^2}, \tag{76}$$

$$g_t^2 = \gamma_t\dot{\gamma}_t - \frac{\dot{\alpha}_t}{\alpha_t}\gamma_t^2 = \gamma_t^2\left(\frac{\dot{\gamma}_t}{\gamma_t} - \frac{\dot{\alpha}_t}{\alpha_t}\right) \tag{77}$$

$$= \gamma^2\frac{d}{dt}\log \frac{\gamma_t}{\alpha_t} \tag{78}$$

$$= \gamma^2\frac{d}{dt}(\frac{1}{2}\log \frac{\sigma_t^2\sigma_1^2}{\sigma_1^2 a_t^2 - \sigma_t^2 a_1^2}) \tag{79}$$

$$= \sigma_t^2\left(1 - \frac{\sigma_t^2 a_1^2}{\sigma_1^2 a_t^2}\right)\left(\frac{\dot{\sigma}_t}{\sigma_t} - \frac{\sigma_1^2 a_t\dot{a}_t - a_1^2\sigma_t\dot{\sigma}_t}{\sigma_1^2 a_t^2 - \sigma_t^2 a_1^2}\right) \tag{80}$$

$$= \frac{\dot{\sigma}_t\sigma_t a_t - \sigma_t^2\dot{a}_t}{a_t}. \tag{81}$$

Therefore,

$$f_t \mathbf{X}_t + s_t \mathbf{x}_T == \bar{\mathbf{f}}_t(\mathbf{X}_t) - \bar{g}_t^2 \bar{\mathbf{h}}_t(\mathbf{X}_t), \bar{g}_t = g_t, \tag{82}$$

which matches the formulation in DDBM.

$\square$

### C.3  EDM

**ODE formulation**. The ODE formulation in EDM can be formlated in our framework as we set $\alpha_t = 1, \beta_t = 0, \gamma_t = \sigma_t$.

*Proof.* Recall 20, the ODE formulation is given by:

$$d\mathbf{X}_t = \left[ f_t \mathbf{X}_t + s_t \mathbf{x}_T - \frac{1}{2} g_t^2 \nabla_{\mathbf{X}_t} \log p_t(\mathbf{X}_t|\mathbf{x}_T) \right] dt \quad \mathbf{X}_T = \mathbf{x}_T \tag{83}$$

where $f_t = \frac{\dot{\alpha}_t}{\alpha_t}, \quad s_t = \dot{\beta}_t - \frac{\dot{\alpha}_t}{\alpha_t}\beta_t, \quad g_t = \sqrt{2(\gamma_t \dot{\gamma}_t - \frac{\dot{\alpha}_t}{\alpha_t}\gamma_t^2)}$. As $\alpha_t = 1, \beta_t = 0, \gamma_t = \sigma_t$, The sampling ODE is given by:

$$d\mathbf{X}_t = -\sigma_t \dot{\sigma}_t \nabla_{\mathbf{x}_t} \log p_t(\mathbf{X}_t) dt \tag{84}$$

$\square$

**Denoising score matching**. The score remarameterization in EDM is the same as ours in Eq. 12. Let $\alpha_t = 1, \beta_t = 0, \gamma_t = \sigma_t$, then the score reparameterization in Eq. 12 is given by:

$$\nabla_{\mathbf{x}_t} \log p_t(\mathbf{X}_t) \approx \frac{\hat{\mathbf{x}}_0 - \mathbf{x}_t}{\sigma_t^2}. \tag{85}$$

**Sampling SDEs with stochasticity added**. Recall Theorem 1, as $\alpha_t = 1, \beta_t = 0, \gamma_t = \sigma_t$, then the SDE has the form:

$$d\mathbf{X}_t = (-\sigma_t \dot{\sigma}_t + \epsilon_t) \nabla_{\mathbf{x}_t} \log p_t(\mathbf{X}_t) dt + \sqrt{2\epsilon_t} d\mathbf{W}_t. \tag{86}$$

Now we recover the stochastic sampling SDE in original EDM paper.

### C.4  I2SB

I2SB can be reformulated in our framework as we let:

$$\alpha_t = 1 - \frac{\sigma_t^2}{\sigma_1^2}, \beta_t = \frac{\sigma_t^2}{\sigma_1^2}, \gamma_t = \sqrt{\sigma_t^2(1 - \frac{\sigma_t^2}{\sigma_1^2})} \tag{87}$$

where $\sigma_t^2 := \int_0^t \beta_\tau d\tau$.

Using discretization 17:

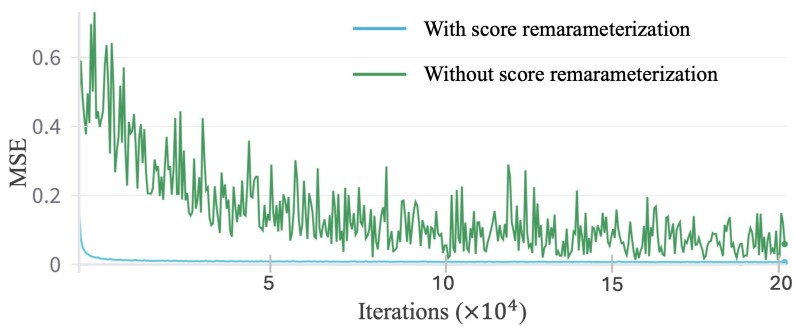

Figure 7: MSEs during training, where $\text{MSE} = \frac{1}{B}\sum_{i=1}^{B}\|\hat{x}_0 - x_0\|^2$.

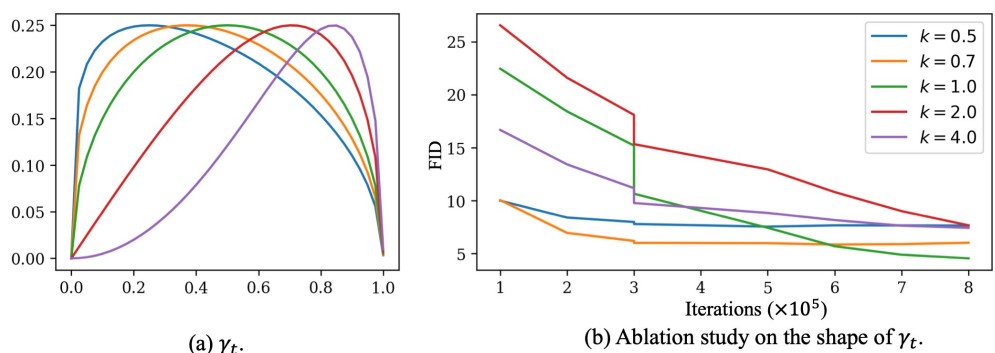

(a) $\gamma_t$.

(b) Ablation study on the shape of $\gamma_t$.

Figure 8: Ablation study on the shape of $\gamma_t$.

$$\mathbf{x}_{t-\Delta t} = (\alpha_{t-\Delta t} - \alpha_t \frac{\beta_{t-\Delta t}}{\beta_t})\hat{\mathbf{x}}_0 + \frac{\beta_{t-\Delta t}}{\beta_t}\mathbf{x}_t + \sqrt{\gamma_{t-\Delta t}^2 - \frac{\beta_{t-\Delta t}^2 \gamma_t^2}{\beta_t^2}}\bar{\mathbf{z}}_t \tag{88}$$

$$= (1 - \frac{\beta_{t-\Delta t}}{\beta_t})\hat{\mathbf{x}}_0 + \frac{\beta_{t-\Delta t}}{\beta_t}\mathbf{x}_t + \sqrt{\gamma_{t-\Delta t}^2 - \frac{\beta_{t-\Delta t}^2 \gamma_t^2}{\beta_t^2}}\bar{\mathbf{z}}_t \tag{89}$$

$$= (1 - \frac{\sigma_{t-\Delta t}^2}{\sigma_t^2})\hat{\mathbf{x}}_0 + \frac{\sigma_{t-\Delta t}^2}{\sigma_t^2}\mathbf{x}_t + \sqrt{\frac{\sigma_{t-\Delta t}^2(1 - \frac{\sigma_{t-\Delta t}^2}{\sigma_1^2})\frac{\sigma_t^4}{\sigma_1^4} - \frac{\sigma_{t-\Delta t}^4}{\sigma_1^4}\sigma_t^2(1 - \frac{\sigma_t^2}{\sigma_1^2})}{\frac{\sigma_t^4}{\sigma_1^4}}}\bar{\mathbf{z}}_t \tag{90}$$

$$= (1 - \frac{\sigma_{t-\Delta t}^2}{\sigma_t^2})\hat{\mathbf{x}}_0 + \frac{\sigma_{t-\Delta t}^2}{\sigma_t^2}\mathbf{x}_t + \sqrt{\frac{\sigma_{t-\Delta t}^2(\sigma_t^2 - \sigma_{t-\Delta t}^2)}{\sigma_t^2}}\bar{\mathbf{z}}_t \tag{91}$$

In the I2SB paper, define $a_n^2 := \int_{t_n}^{t_{n+1}} \beta_\tau d\tau$, $\sigma_n^2 := \int_0^{t_n} \beta_\tau d\tau$. Therefore,

$$\mathbf{x}_n = \frac{a_n^2}{a_n^2 + \sigma_n^2}\hat{\mathbf{x}}_0 + \frac{\sigma_n^2}{a_n^2 + \sigma_n^2}\mathbf{x}_{n+1} + \sqrt{\frac{\sigma_n^2 a_n^2}{\alpha_n^2 + \sigma_n^2}}\bar{\mathbf{z}}_t \tag{92}$$

Thus, we reproduce the sampler of I2SB.

## D  ADDITIONAL DESIGN GUIDELINE

**Score reparameterization.** We compared the training stability with and without score reparameterization using the DIODE ($64 \times 64$) dataset, and the results are shown in Fig. 7. For training without

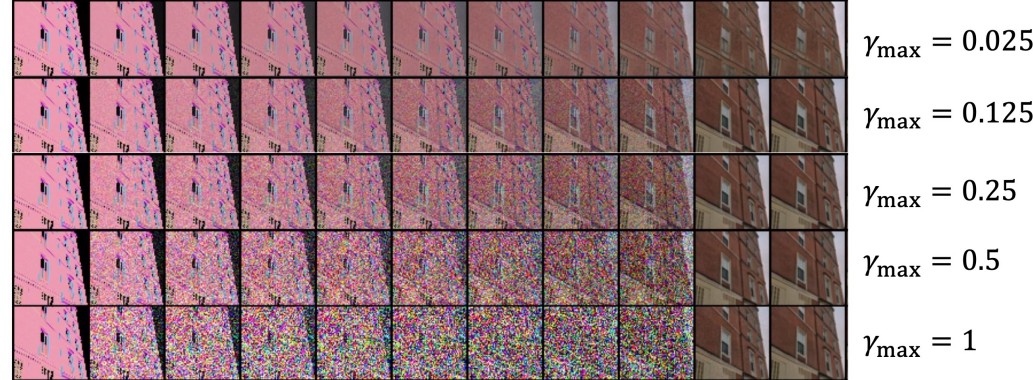

Figure 9: Sampling paths with dfferent choices of $\gamma_t$. As $\gamma_t$ extreamly low, e.g, $\gamma_{\max} = 0.025$, the model will be failed to construct details of images.

score reparameterization, the score function $s_\theta(\mathbf{x}, \mathbf{x}_T, t)$ is parameterized by a neural network, and $\hat{\mathbf{x}}_0(\mathbf{x}, \mathbf{x}_T, t)$ is computed as: $\hat{\mathbf{x}}_0(\mathbf{x}, \mathbf{x}_T, t) = \frac{1}{\alpha_t}\left(\gamma_t^2 s_\theta(\mathbf{x}, \mathbf{x}_T, t) + \mathbf{x}_t - \beta\mathbf{x}_T\right)$. For training with score reparameterization, $\hat{\mathbf{x}}_0(\mathbf{x}, \mathbf{x}_T, t)$ is directly parameterized as a neural network. We then compared the mean squared error (MSE) between $\hat{\mathbf{x}}_0$ and $\mathbf{x}_0$ during training. The results in Fig. 7 indicate that score reparameterization helps reduce training instability.

$\alpha_t$ **and** $\beta_t$. Theoretically, $\alpha_t$ and $\beta_t$ can be freely designed, and future work may explore alternative design choices. However, in this paper, we focus on the simple case where $\alpha_t = 1 - t$ and $\beta_t = t$. The rationale is as follows: consider the scenario where $\alpha_t = 1 - \beta_t$, which represents an interpolation along the line segment between $x_0$ and $x_1$. For the path $p_t^{(1)}(x) = \mathcal{N}((1 - \beta_t)x_0 + \beta_t x_1, \gamma_t^2 \mathbf{I})$, where $\beta_t$ is invertible, it is straightforward to construct another path $p_t^{(2)}(x) = \mathcal{N}((1 - t)x_0 + tx_1, \gamma_{\beta_t^{-1}}^2 \mathbf{I})$, which achieves the same objective function but uses a different distribution of $t$ during training. Based on this equivalence, setting $\alpha_t = 1 - t$ and $\beta_t = t$ is a reasonable choice.

**The shape of** $\gamma_t$. We conducted an ablation study on $\gamma_t$ with different shapes. Specifically, we assumed $\gamma_t$ has the form $\gamma_t = 2\gamma_{\max}\sqrt{t^k(1 - t^k)}$, as shown in Fig. 8, $\gamma_t$ will have different shape as we set different $k$. The results indicate that the best performance is achieved when $k = 1$, which is the exact setting used in this paper.

$\gamma_{\max}$. Our ablation studies on $\gamma_{\max}$ demonstrate that the optimal values of $\gamma_{\max}$ are approximately $0.125$ or $0.25$. Furthermore, the sampling paths corresponding to different choices of $\gamma_t$ are shown in Fig. 9. Adding an appropriate amount of noise to the transition kernel helps in constructing finer details.

$\epsilon_t$. We use the setting $\epsilon_t = \eta\left(\gamma_t\dot{\gamma}_t - \frac{\dot{\alpha}_t}{\alpha_t}\gamma_t^2\right)$. The ablation studies on $\epsilon_t$ demonstrate that the optimal choice of $\eta$ for the DDBM-VP model is approximately $0.3$, while the best choice for the SDB model with a Linear Path is around $1.0$. Additionally, we present sample paths and generated images under different $\eta$ settings to illustrate heuristic parameter tuning techniques. The results are shown in Figures 11, 12, and 13. Too small a value of $\eta$ results in the loss of high-frequency information, while too large a value of $\eta$ produces over-sharpened and potentially noisy sampled images.

# E  EXPERIMENT DETAILS

**Architecture**. We maintain the architecture and parameter settings consistent with (Linqi Zhou et al., 2023), utilizing the ADM model (Dhariwal & Nichol, 2021) for $64 \times 64$ resolution, modifying the channel dimensions from 192 to 256 and reducing the number of residual blocks from three to two. Apart from these changes, all other settings remain identical to those used for $64 \times 64$ resolution.

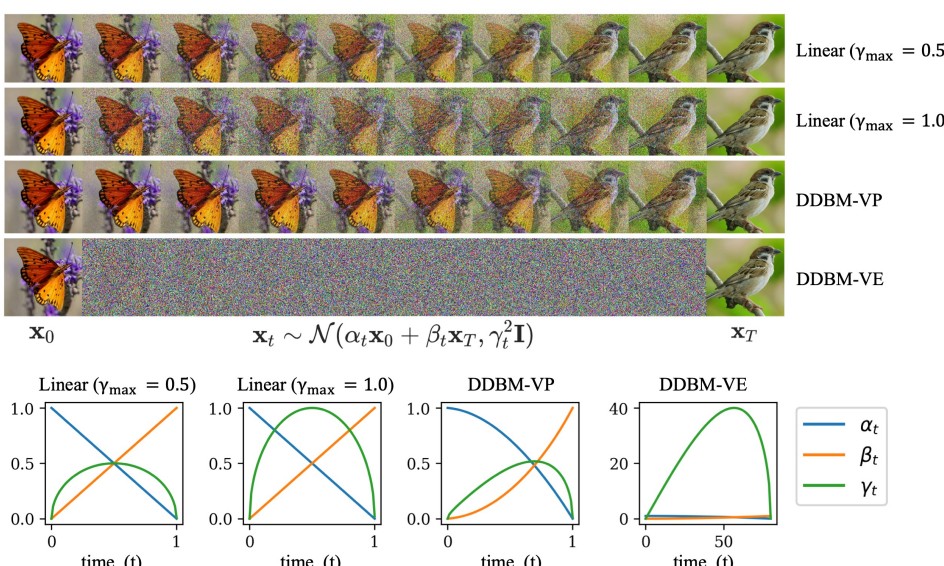

Figure 10: An illustration of design choices of transition kernels and how they affect the I2I translation process. $\alpha_t$ and $\beta_t$ define the interpolation between two images, while $\gamma_t$ controls the noise added to the process. ntuitively, the DDBM-VE model introduces excessive noise in the middle stages, which is unnecessary for effective image translation and may explain its poor performance. In contrast, our Linear path results in a symmetrical noise schedule, ensuring a more balanced process. On the other hand, the DDBM-VP path adds more noise near $\mathbf{x}_T$, , indicating that during training, more computational resources are focused around $\mathbf{x}_0$.

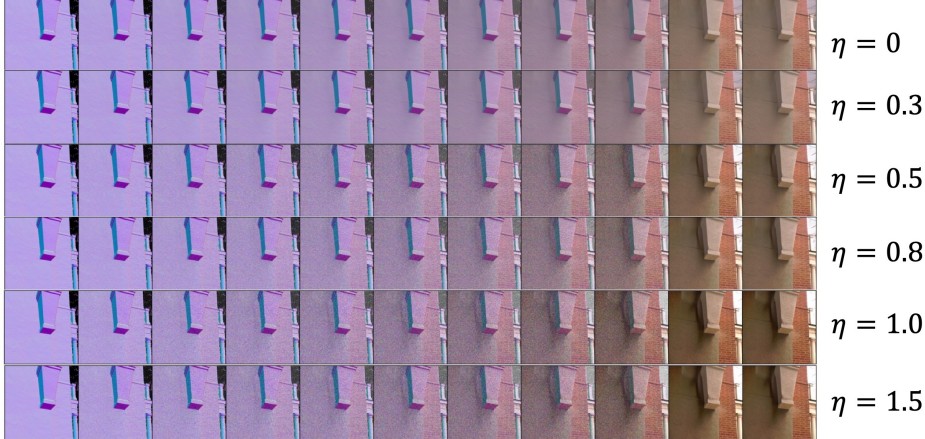

Figure 11: Sampling path with dfferent choices of $\epsilon_t$. As $\epsilon_t = 0$, the generated images lack details, as $\epsilon_t$ too large, the sampled images are over-sharpening. The best choices of $\epsilon_t$ are around $\epsilon_t = 0.8$ and $\epsilon_t = 1.0$.

**Training**. We include additional pre- and post-processing steps: scaling functions and loss weighting, the same ingredient as (Karras et al., 2022). Let $D_\theta(\mathbf{x}_t, \mathbf{x}_T, t) = c_{\text{skip}}(t)\mathbf{x}_t + c_{\text{out}(t)}(t)F_\theta(c_{\text{in}}(t)\mathbf{x}_t, c_{\text{noise}}(t))$, where $F_\theta$ is a neural network with parameter $\theta$, the effective training target with respect to the raw network $F_\theta$ is: $\mathbb{E}_{\mathbf{x}_t, \mathbf{x}_0, \mathbf{x}_T, t}\left[\lambda \| c_{\text{skip}}(\mathbf{x}_t + c_{\text{out}}F_\theta(c_{\text{in}}\mathbf{x}_t, c_{\text{noise}}) - \mathbf{x}_0\|^2\right]$. Scaling scheme are chosen by requiring network inputs and training targets to have unit variance ($c_{\text{in}}, c_{\text{out}}$), and amplifying errors in $F_\theta$ as little as possible. Following reasoning in (Linqi Zhou et al., 2023),

$$c_{\text{in}}(t) = \frac{1}{\sqrt{\alpha_t^2 \sigma_0^2 + \beta_t^2 \sigma_T^2 + 2\alpha_t \beta_t \sigma_{0T} + \gamma_t^2}}, \quad c_{\text{skip}}(t) = (\alpha_t \sigma_0^2 + \beta_t \sigma_{0T}) * c_{\text{in}}^2, \quad (93)$$

$$c_{\text{out}}(t) = \sqrt{\beta_t^2 \sigma_0^2 \sigma_1^2 - \beta_t^2 \sigma_{0T}^2 + \gamma_t^2 \sigma_0^2} c_{\text{in}}, \quad \lambda = \frac{1}{c_{\text{out}}^2}, \quad c_{\text{noise}}(t) = \frac{1}{4}\log(t), \quad (94)$$

where $\sigma_0^2, \sigma_T^2$, and $\sigma_{0T}$ denote the variance of $\mathbf{x}_0$, variance of $\mathbf{x}_T$ and the covariance of the two, respectively.

We note that TrigFlow (Lu & Song, 2024), a contemporaneous work, adopts the same score reparameterization and pre-conditioning techniques. It can be considered a special case of our framework by setting $\alpha_t = \cos(t), \beta_t = 0, \gamma_t = \sigma_0 \sin(t), t \in [0, \frac{\pi}{2}]$. In this case, $\sigma_T = 0, \sigma_{0T} = 0$,

$$c_{\text{in}}(t) = \frac{1}{\sqrt{\alpha_t^2 \sigma_0^2 + \gamma_t^2}} = \frac{1}{\sqrt{\sin^2(t)\sigma_0^2 + \cos^2(t)\sigma_0^2}} = \frac{1}{\sigma_0}, \quad (95)$$

$$c_{\text{skip}}(t) = (\alpha_t \sigma_0^2)c_{in}^2 = \cos(t) \cdot \sigma_0^2 \cdot \frac{1}{\sigma_0^2} = \cos(t), \quad (96)$$

$$c_{out}(t) = \sqrt{\gamma_t^2 \sigma_0^2} \cdot c_{in} = \sin(t)\sigma_0, \quad (97)$$

$$D_\theta(x_t, t) = c_{\text{skip}}x_t + c_{\text{out}}F_\theta(c_{\text{in}}x_t, c_{\text{noise}}) = \cos(t)x_t + \sin(t)\sigma_0 F_\theta(\frac{1}{\sigma_0}, c_{\text{noise}}). \quad (98)$$

Then we recover TrigFlow.

In our implementation, we set $\sigma_0 = \sigma_T = 0.5, \sigma_{0T} = \sigma_0^2/2$ for all training sessions. Other setting are shown in Table 7.

Table 7: Training settings

| | Dataset | edges→handbags | edges→handbags | edges→handbags |
|---|---|---|---|---|
| Model | $\eta$ | 0 | 0 | 0.5 |
| | $\gamma_{\max}$ | 0.125 | 0.25 | 0.125 |
| Setting | GPU | 1 A6000 48G | 1 H100 96G | 1 H100 96G |
| | Batch size | 32 | 128 | 200 |
| | Learning rate | $1 \times 10^{-5}$ | $5 \times 10^{-5}$ | $1 \times 10^{-4}$ |
| | epochs | 2078 | 2106 | 1443 |
| | Training time | 42 days | 8 days | 11 days |
| | Dataset | DIODE ($256 \times 256$) | DOIDE ($256 \times 256$) | |
| Model | $\eta$ | 0 | 0 | |
| | $\gamma_{\max}$ | 0.125 | 0.25 | |
| Setting | GPU | 1 H100 96G | 1 H100 96G | |
| | Batch size | 16 | 16 | |
| | Learning rate | $2 \times 10^{-5}$ | $2 \times 10^{-5}$ | |
| | epochs | 2617 | 1745 | |
| | Training time | 17 days | 25 days | |

**Sampling**. We use the same timesteps distributed according to EDM (Karras et al., 2022): $(t_{\max}^{1/\rho} + \frac{i}{N}(t_{\min}^{1/\rho} - t_{\max}^{1/\rho}))^\rho$, where $t_{\min} = 0.001$ and $t_{\max} = 1 - 10^{-4}$. The best performance achieved by setting $\rho = 0.6$ for Edges2handbags and $\rho = 0.8$ for DIODE datasets.

**Licenses**

- Edges→Handbags Isola et al. (2017): BSD license.
- DIODE-Outdoor Vasiljevic et al. (2019): MIT license.

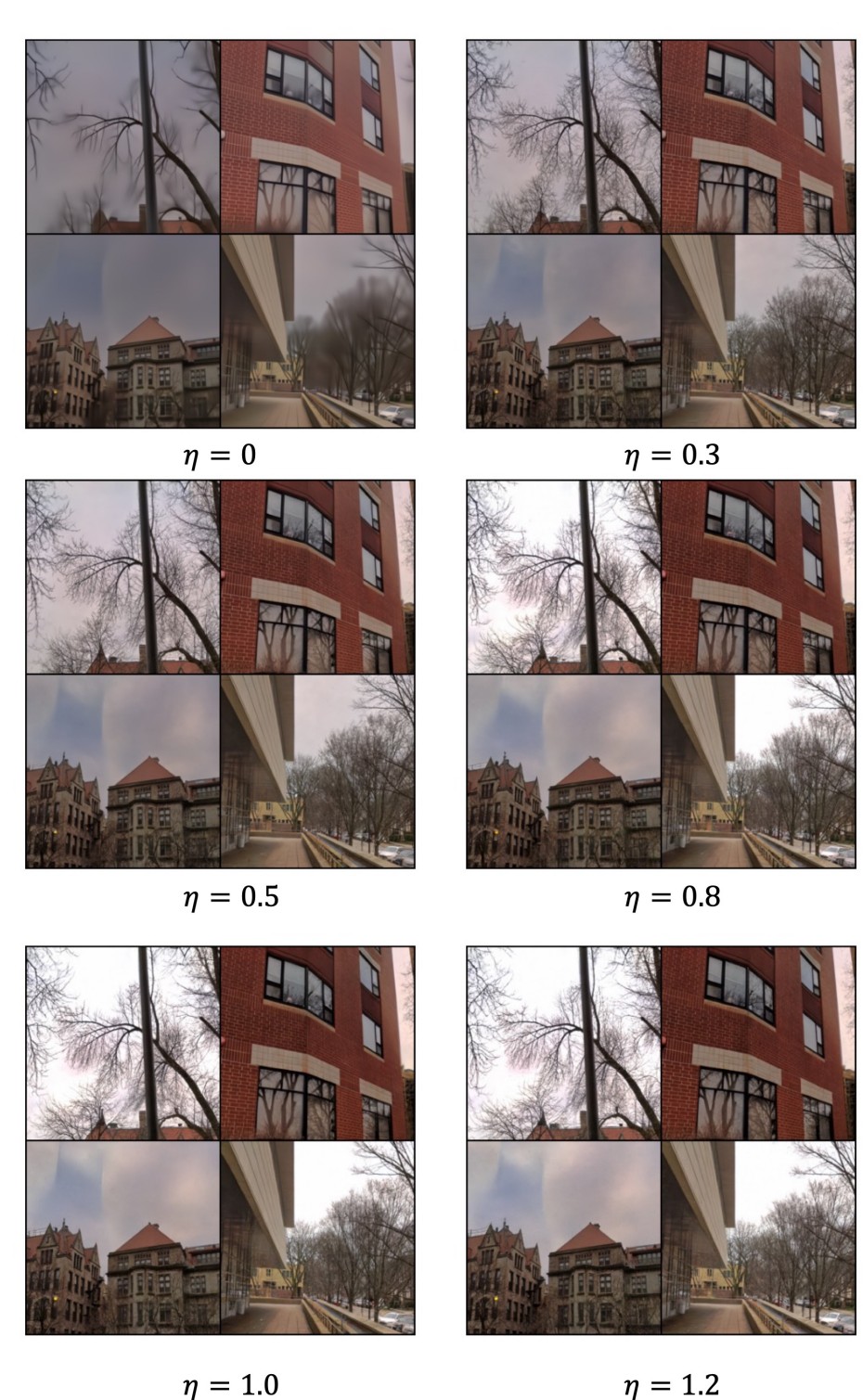

Figure 12: Comparison of sampled images with different $\epsilon_t$ for SDB model, where $\epsilon_t = \eta(\gamma_t \dot{\gamma}_t - \frac{\dot{\alpha}_t}{\alpha_t}\gamma_t^2)$, $\gamma_{\max} = 0.25$, $b = 0$.

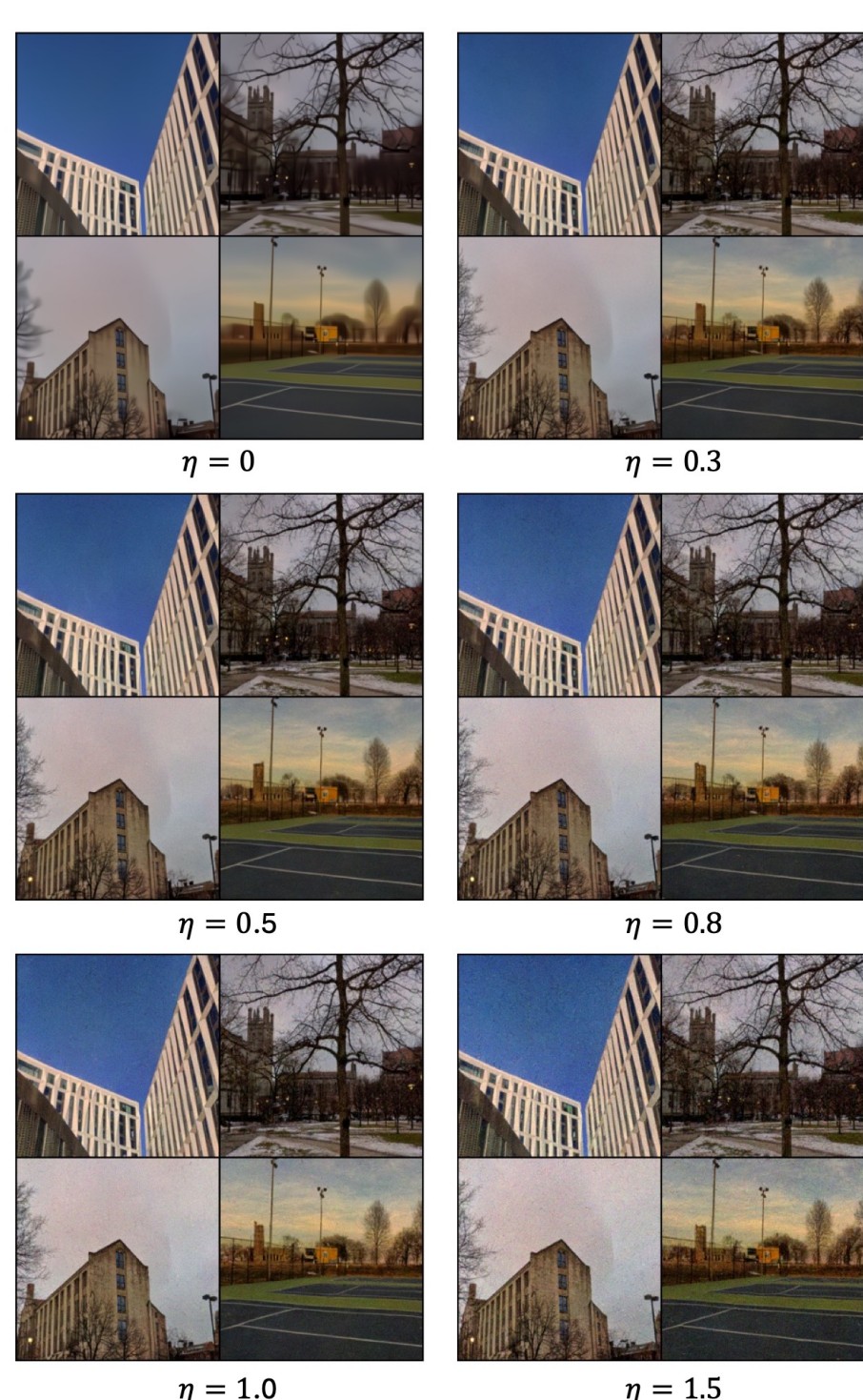

Figure 13: Comparison of sampled images with different $\epsilon_t$ for DDBM-VP pretrained model, where $\epsilon_t = \eta(\gamma_t \dot{\gamma}_t - \frac{\dot{\alpha}_t}{\alpha_t}\gamma_t^2)$.

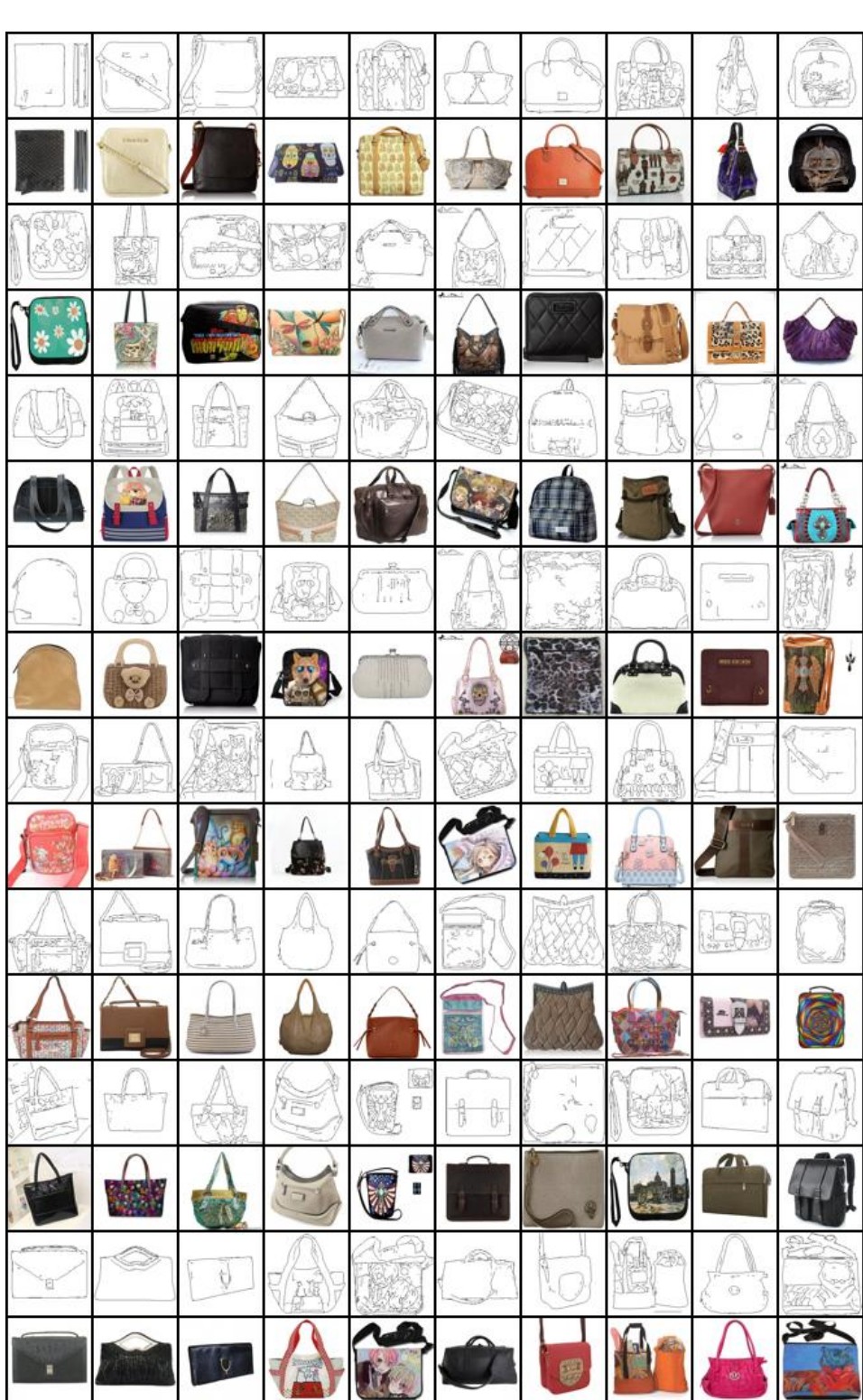

Figure 14: SDB model and sampler ( $\gamma_{\max} = 0.125$, $\eta = 1$, $b = 0$, NFE=5, FID=0.89).

## F  ADDITIONAL VISUALIZATIONS

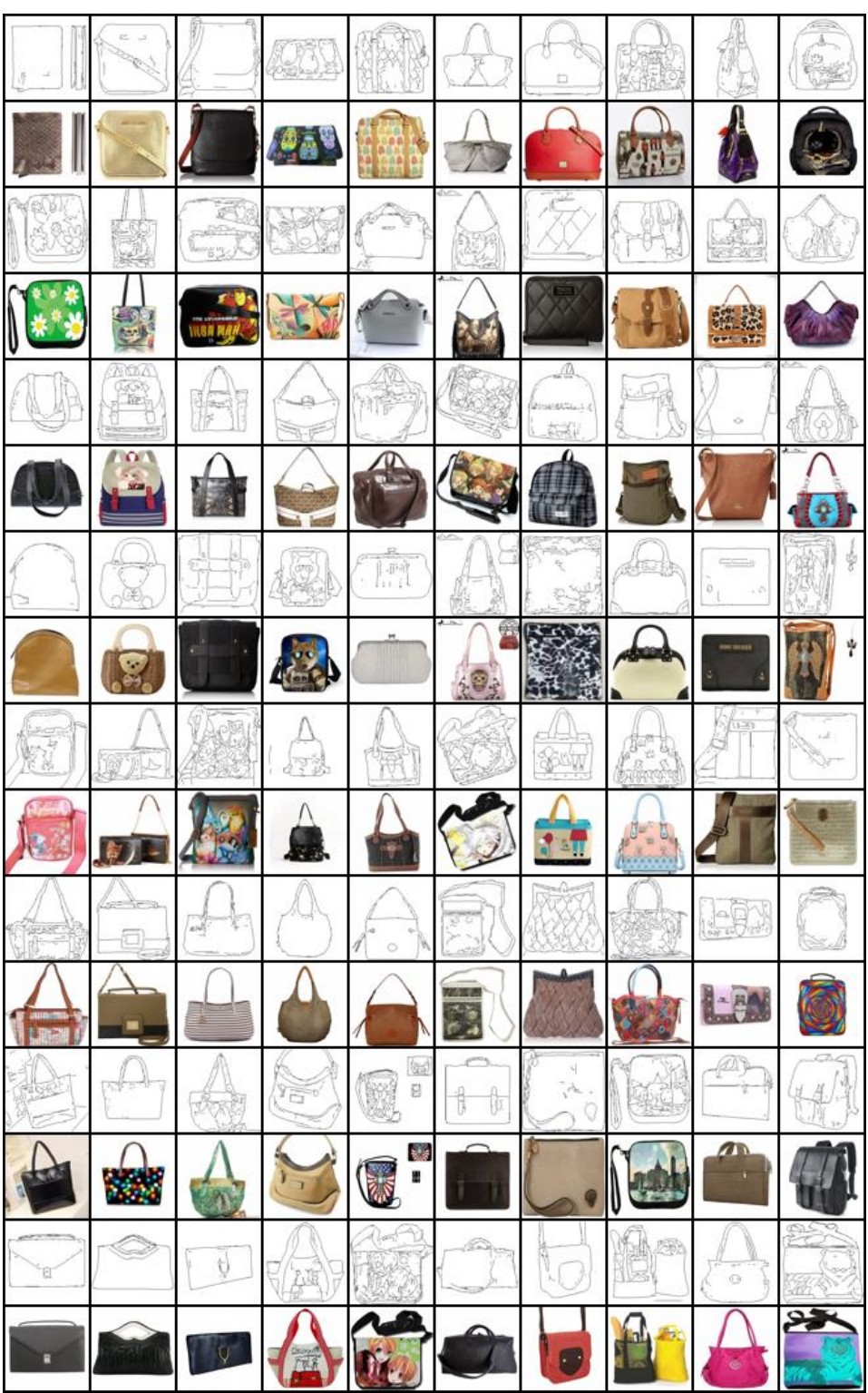

Figure 15: DDBM model and Our sampler (NFE=20, FID=1.53).

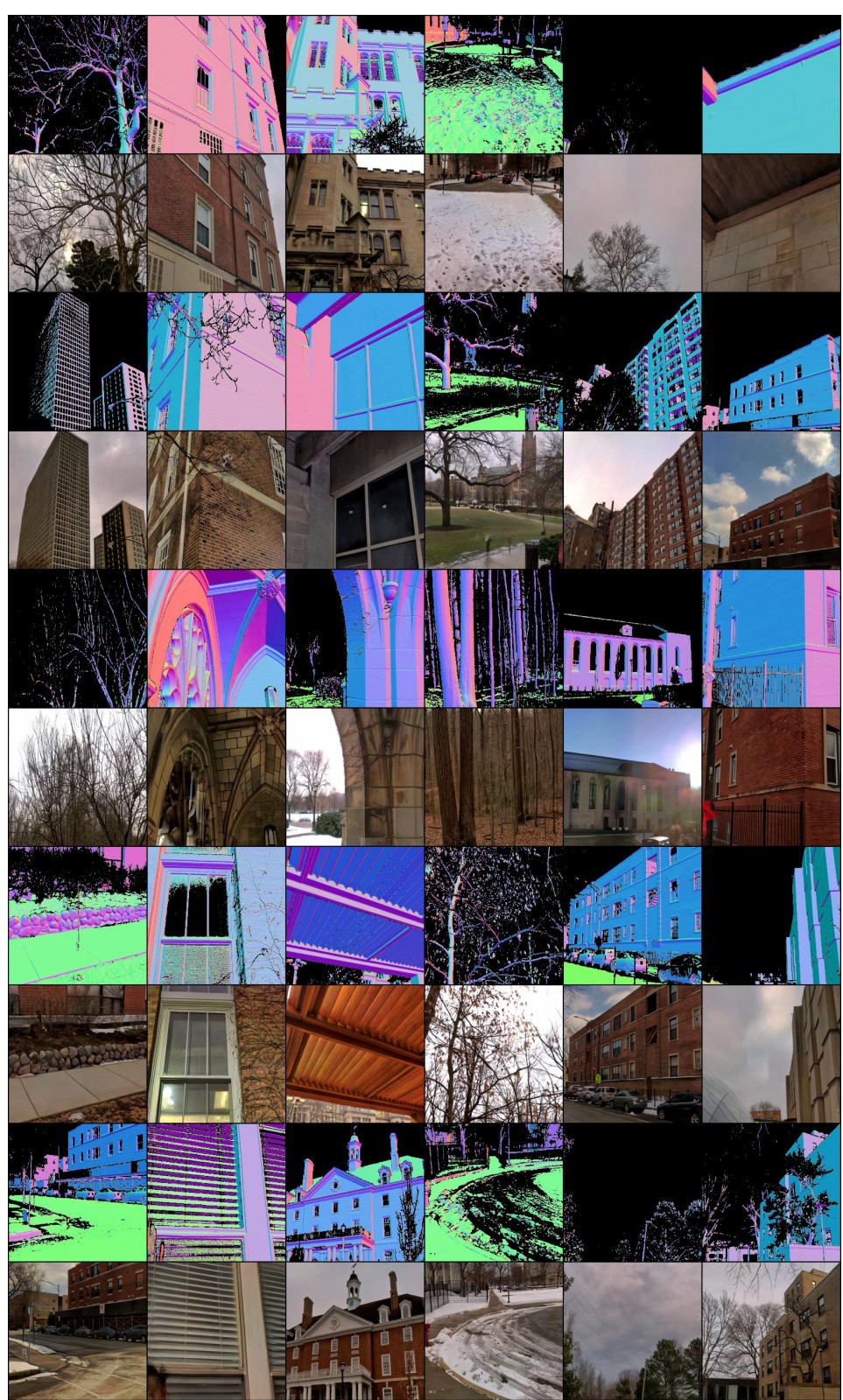

Figure 16: DDBM model and SDB sampler ($\eta = 0.3$, NFE=20, FID=4.12). Samples for DIODE dataset (conditoned on depth images).

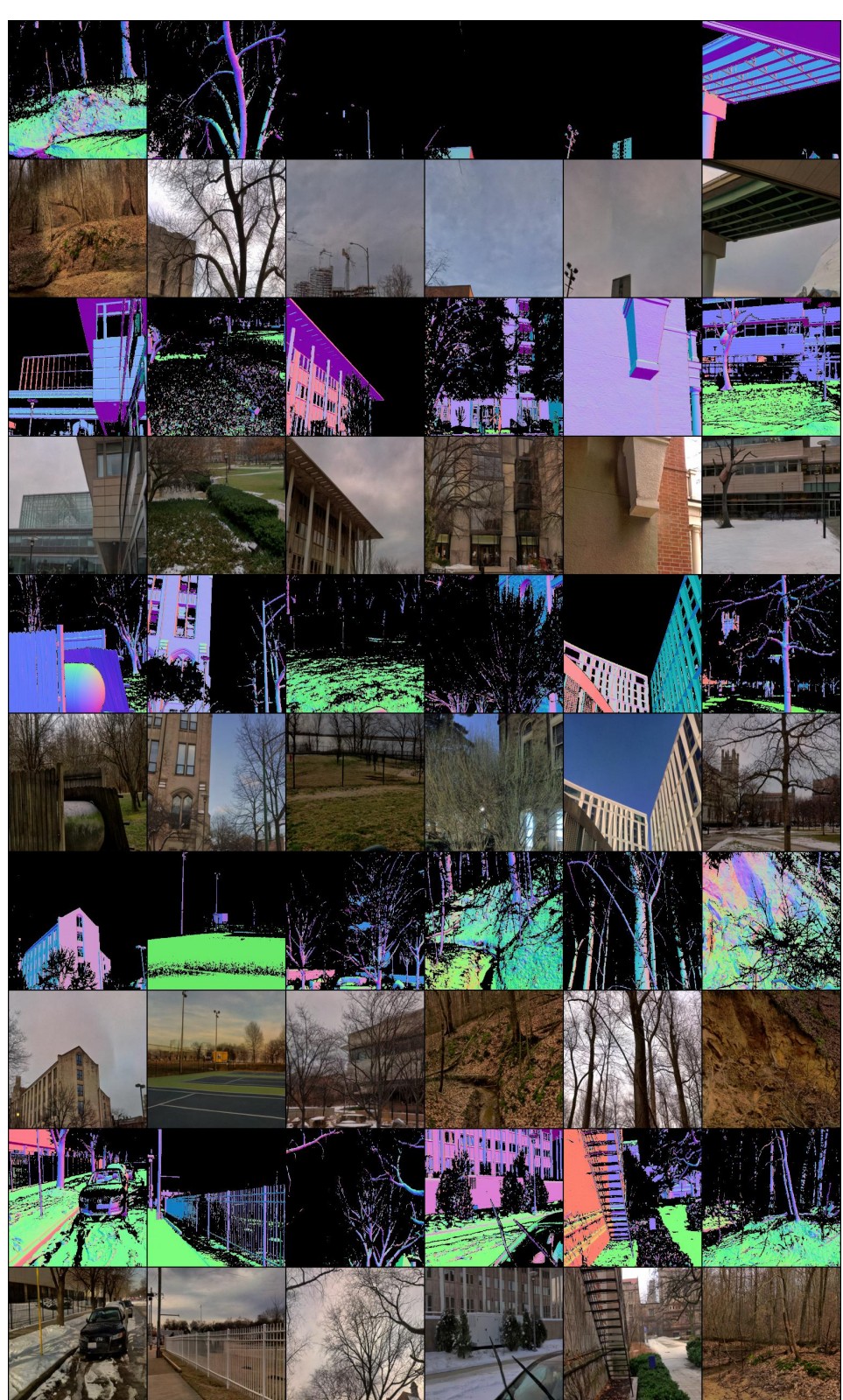

Figure 17: SDB model and sampler ($\gamma_{\max} = 0.25$, $\eta = 1.0$, $b = 0$, NFE=5, FID = 4.16).

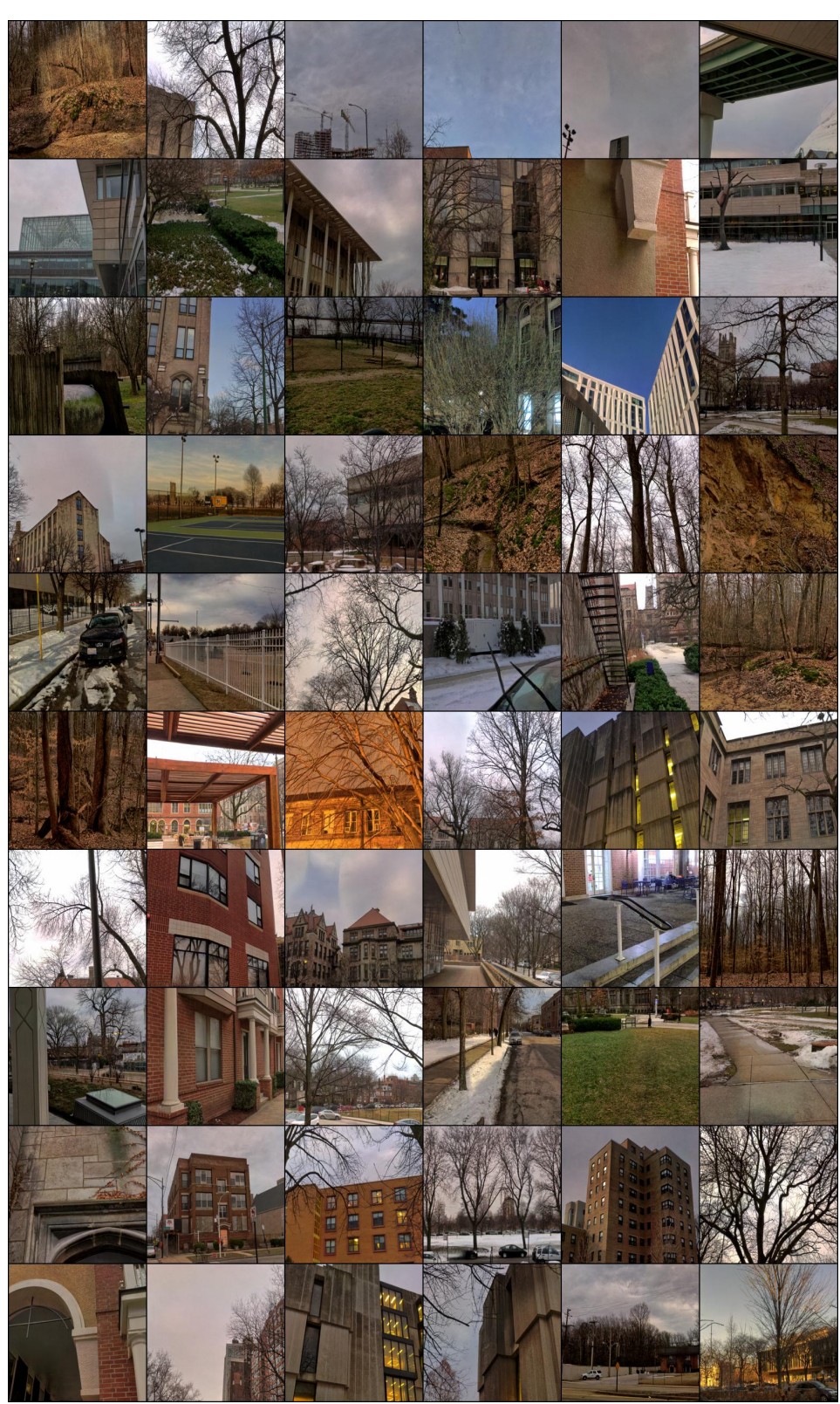

Figure 18: SDB model and sampler ($\gamma_{\max} = 0.25$, $\eta = 1.0$, $b = 0$, NFE=20, FID = 3.27).

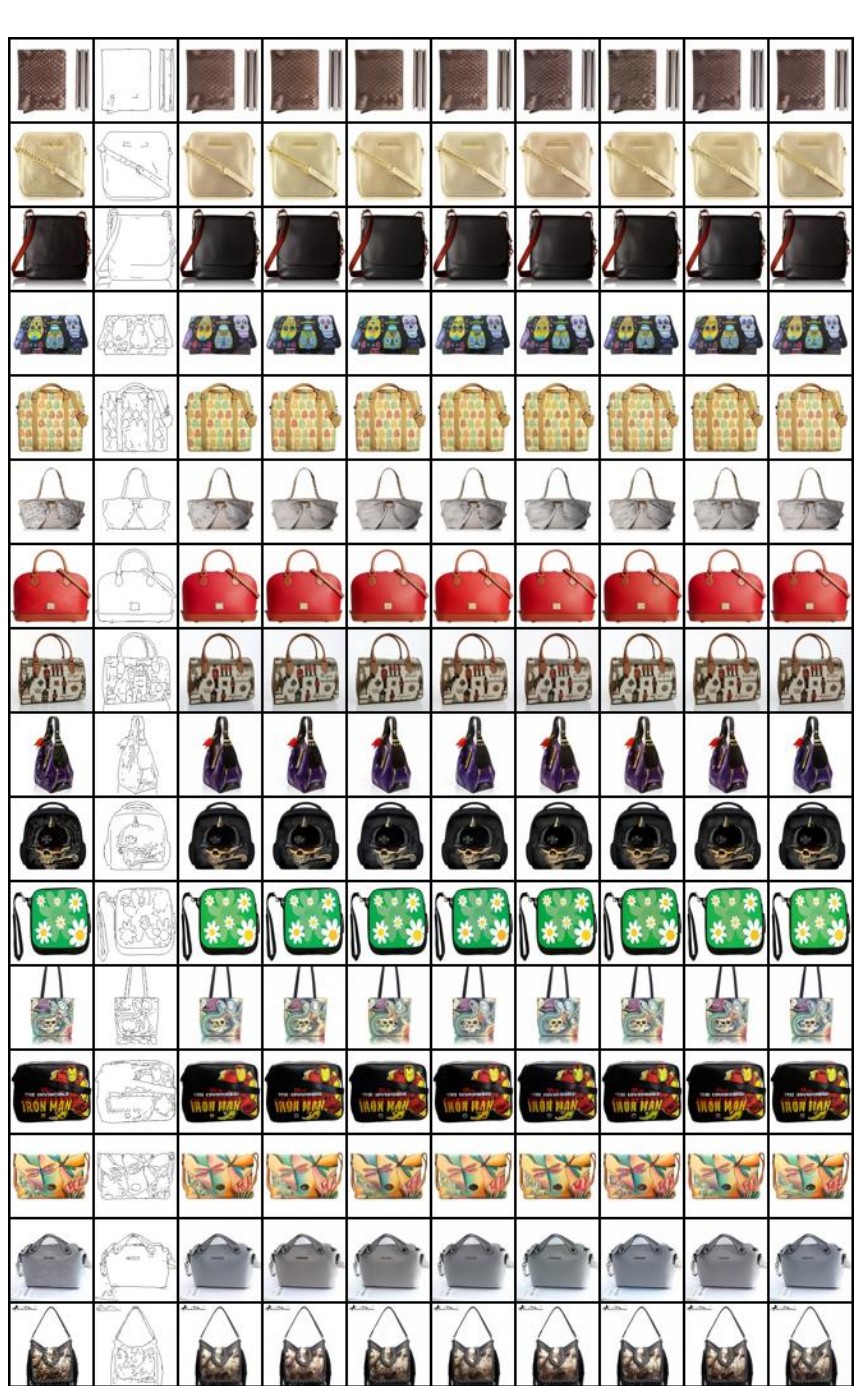

Figure 19: DDBM model and DBIM sampler (NFE=10, FID = 2.46, AFD=5.20).

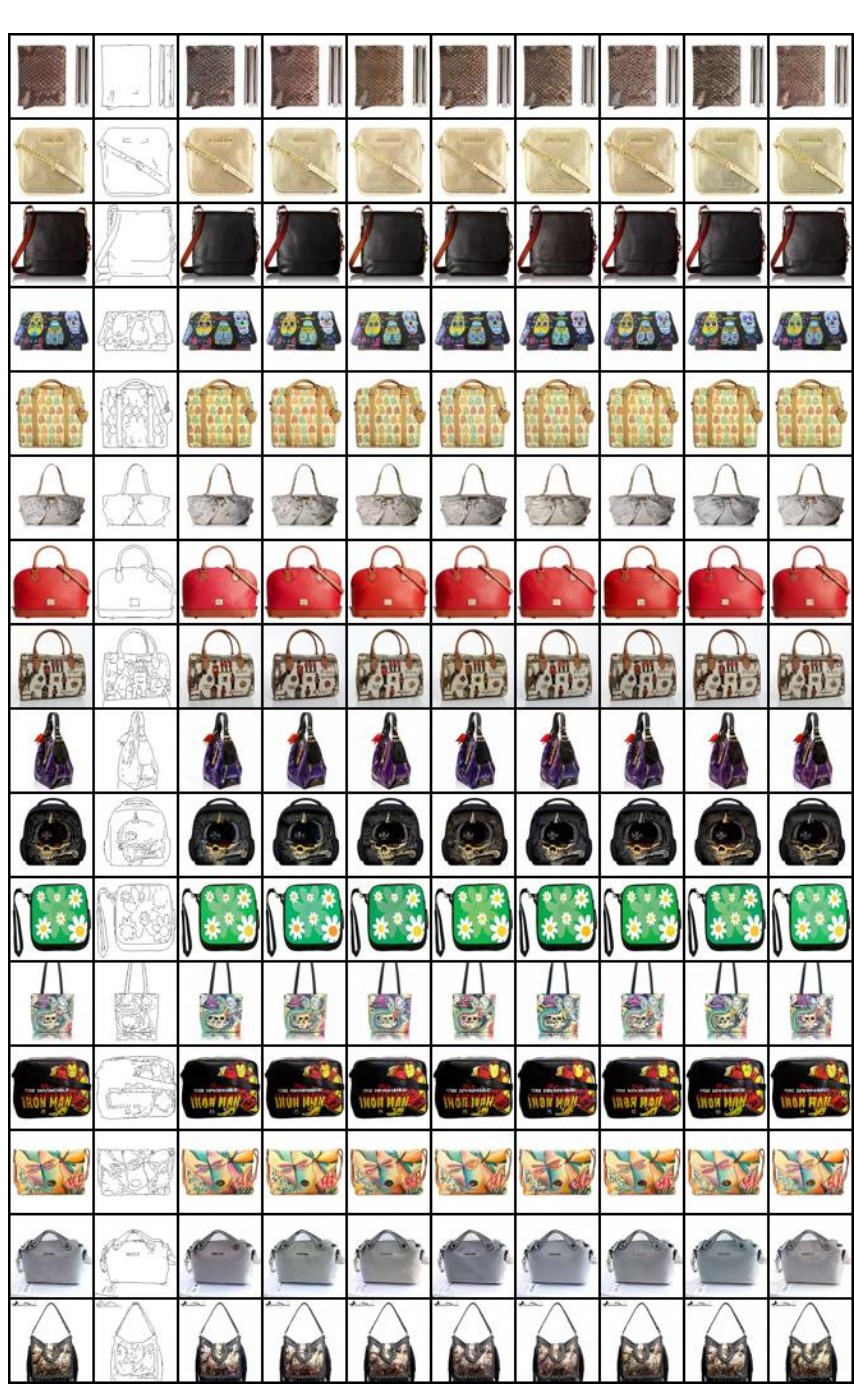

Figure 20: DDBM model and sampler (NFE=118, FID = 1.83, AFD=6.99).

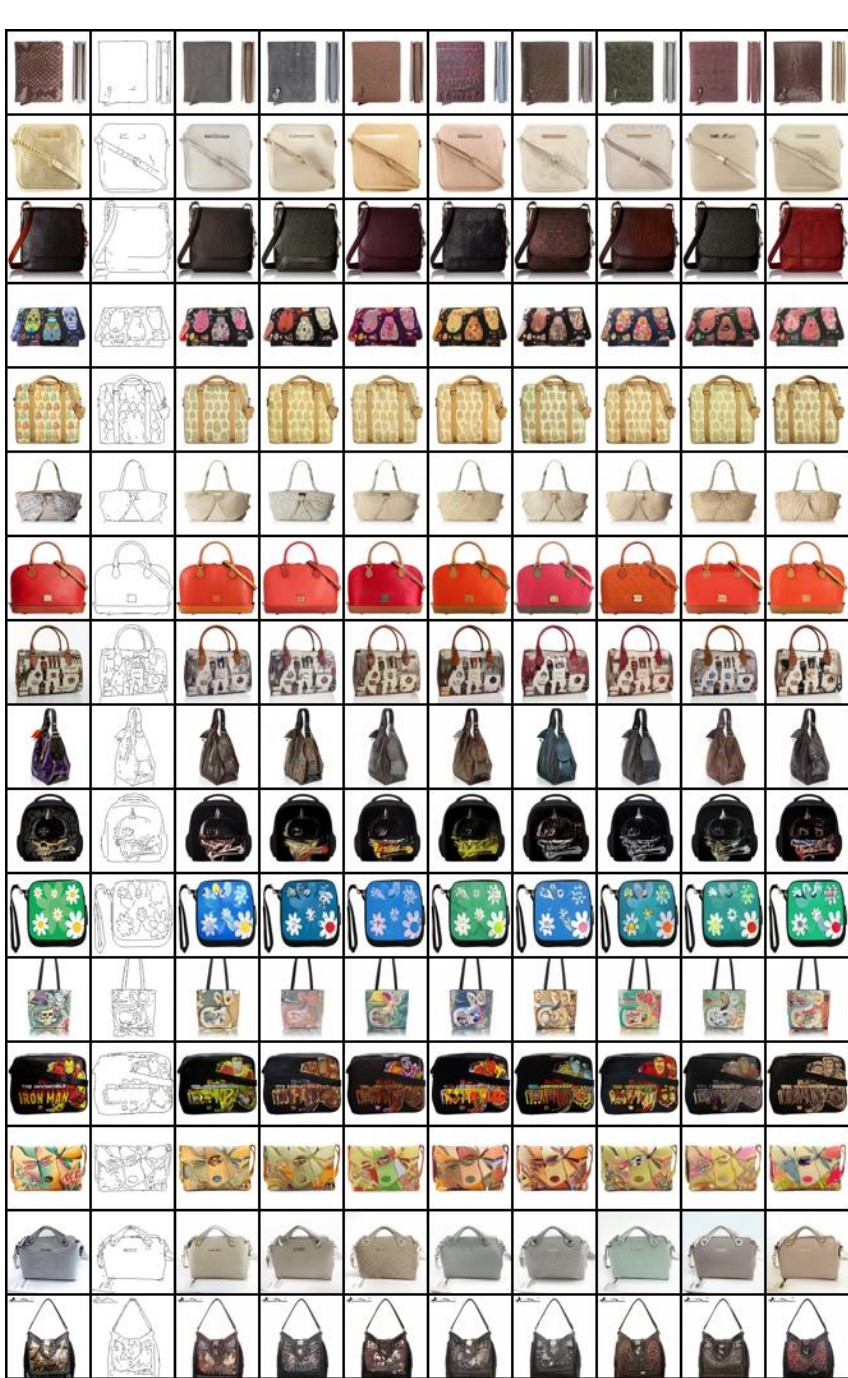

Figure 21: SDB model and sampler ($\gamma_{\max} = 0.125$, $b = 1.0$, NFE=10, FID = 2.07, AFD=9.35).

