# OpenReview forum: "Exploring the Design Space of Diffusion Bridge Models via Stochasticity Control"
_ICLR.cc/2025/Conference — Submitted to ICLR 2025_

### Official Review · Reviewer_o3pD · 2024-11-01

**Soundness:** 3
**Presentation:** 2
**Contribution:** 2
**Rating:** 6
**Confidence:** 2

**Summary:**

This paper proposes the Stochastic-controlled Diffusion Bridge (SDB) model to improve the effect of noise generated by SDEs, transition kernel, and base distribution on sampling when the existing Diffusion Bridge models (DBMs) are sampling image-to-image translation. The authors introduce stochasticity control (SC) in three areas—sampling stochastic differential equations (SDEs), transition kernels, and base distributions—to improve sampling efficiency, image quality, and diversity in output. In this process, the paper presents a metric for measuring the diversity of output. The SC mechanism mitigates singularities during training and sampling, and introduces flexibility in design, allowing SDB to outperform existing models like DDBM and DBIM on benchmark tasks, achieving lower FID scores, faster sampling, and improved image diversity.

**Strengths:**

**1. Existing Model Shortcomings: Identification and Solutions**
Recognize the limitations of existing diffusion bridge models as stemming from noise introduced during image-to-image translation sampling. The SC mechanism effectively addresses these shortcomings by introducing enhanced flexibility and precise control over the noise at various stages of the diffusion process, thereby mitigating the inherent limitations of conventional diffusion models.

**2. Experimental Validation**
The paper provides extensive quantitative comparisons with multiple baseline models, demonstrating superior performance in FID, sampling efficiency, and conditional diversity on I2I tasks.

**3. Expression Reconfiguration**
The paper introduces score reparameterization and discretization schemes to address stability issues in previous diffusion bridge methods, offering novel insights into stochastic diffusion processes.

**Weaknesses:**

**1. Limited Discussion on Computational Overhead**
Although the model achieves lower FID scores with fewer function evaluations, the added complexity of SC mechanisms (especially in tuning multiple noise schedules) may increase computational overhead. A direct comparison of computational resources (e.g., time or memory) required by SDB versus other DBMs would provide a clearer picture of the trade-offs.

**2. Lack of Detailed Analysis on the Contribution of Each Stochasticity Control Component**
While the authors claim that SC enhances image quality, sampling efficiency, and conditional diversity, there is a lack of detailed analysis on how each component (e.g., SC in the transition kernel, SC in the base distribution) individually contributes to these improvements. Although overall performance gains are evident, the authors have divided Stochasticity Control into three distinct components, yet the extent to which each component contributes to the model's improvement is not adequately demonstrated through ablation studies.

**3. Lack of Clear Design Guidelines for Stochasticity Control**
The paper introduces various strategies for enhancing performance through Stochasticity Control (SC). However, it lacks clear principles for determining optimal parameters in SC design. For instance, the design of the ϵt value used in SC is largely determined empirically, which complicates the systematic selection of SC parameters suitable for specific tasks. Additionally, the proposed method includes several hyperparameters (e.g.,  αt, βt, γt, ϵt), yet there is no concrete guideline on how these should be chosen. While certain values were empirically set during experiments, it remains unclear how these can be applied in more general cases. To address this, it would be helpful if the authors could provide a sensitivity analysis for these key hyperparameters, showing how performance varies across different settings. Additionally, we suggest the authors share any rules of thumb or heuristics they relied on when selecting these parameters, which would provide practical guidance for applying the method in various contexts.

**4. Lack of Reliability for the Proposed Evaluation Metric**
The authors propose Average Feature Distance (AFD) as a metric for assessing diversity. However, there is insufficient analysis to establish the reliability and validity of this metric. To substantiate AFD as a credible measure of diversity, the authors should have demonstrated its application across multiple existing baselines and metrics, thereby enhancing its reliability. In particular, there is a lack of comparative analysis and explanation on how AFD interacts with other evaluation metrics and effectively represents trustworthy conditional diversity. To establish the reliability of AFD, it would be beneficial for the authors to compare AFD scores with human judgments of diversity or apply AFD to datasets with known diversity characteristics. Additionally, case studies illustrating how AFD addresses insights that are missed by similar existing metrics would strengthen its credibility as a measure of diversity.

**Questions:**

**1. Lack of Theoretical Explanation**
The various concepts and mechanisms introduced in the paper, such as Stochasticity Control and Score Reparameterization, are not sufficiently clear or intuitively explained. While the mathematical descriptions provide a theoretical foundation, the paper would benefit from a figure or diagram that illustrates how Stochasticity Control interacts with other components of the model. Including a main figure to depict the overall framework and interconnections within the proposed model would greatly enhance readers' understanding of the core concepts.

**2. Purpose of Figure 1**
Figure 1 aims to provide an intuitive illustration of how the transition kernels in image-to-image (I2I) translation can incorporate noise through the configuration of multiple hyperparameters, as proposed by the authors. However, since the ultimate goal of I2I translation is to transform the source image into the target image, evaluating only the final translated result may suffice. Figure 1 merely demonstrates the progressive addition of noise, making it challenging to discern the specific impact of these parameters. A clearer explanation of the intended purpose of Figure 1 would be beneficial.

I would be willing to consider increasing the rate if clear responses are provided to the points raised in the Weakness and Question sections.

---

> ### Author Response · Authors · 2024-11-19
> **Reply to reviewer o3pD**
>
> **Limited Discussion on Computational Overhead.**
>
> - In our experiments, we observed that the function evaluation step during each sampling cycle is the most computationally intensive, taking 241.598 ms, while the remaining operations require only 9.343 ms (measured on an NVIDIA H100 GPU with a batch size of 200 for the edges2handbags generation task). This demonstrates that function evaluation is the dominant factor in computational cost, which is why we rely on the number of function evaluations (NFE) as a key metric for comparing computational loads across models.
> - Compared to DDBM, I2SB, and DBIM, our model employs much simpler transition kernels, characterized by linear linear $\alpha_t$ and $\beta_t$, and quadratic $\gamma_t^2$, which should brings simplification in the calculation. Still, we claim that the improvement is trivial, most of the computing resources are spent on function evaluations.
>
> **Lack of Detailed Analysis on the Contribution of Each Stochasticity Control Component**
>
> -  **Ablation studies on sampling algorithms, $\gamma_{\max}$, and $\epsilon_t$**. These results are presented in Section 4 (Experiments) and illustrated in Figure 3.
> -  **Additional evaluation on more dataset and $\gamma_{\max}$**. We include additional experiments with $\gamma_{\max}=0.25$ and DIODE ($256 \times 256$) dataset, as shown in Table 3.
> - **Verification of Reparameterization**. We include new experiments analyzing the training stability brought by the proposed reparameterization. These results, including insights on numerical stability, are shown in Figure 7 in Appendix D.
> -  **Enhanced diversity evaluation**. In Table 4, we added more evaluation on different samplers and different choices of $b$.
>
> **Lack of Clear Design Guidelines for Stochasticity Control**
>
> - **Guidelines based on ablation studies**. Our ablation study on $\gamma_{max}$ demonstrates that the best choices are around $\gamma_{max}=0.125$ and $\gamma_{max}=0.25$, this results further evaluated through Edges2handbags and DIODE ($256 \times 256$) datasets, as shown in Table 3. Our ablation study on $\epsilon_t$ shows that the best choices are around $0.8$ and $1.0$ for different Linear models, even for different $\gamma_{\max}$ and $b$.
>
> - **Additional heuristic guidelines**. We included more design guidelines in the Appendix D. We have also added discussions on the selection of $\alpha_t$, $\beta_t$, the shape of $\gamma_t$, $\gamma_{\max}$ and $\epsilon_t$, which potentially helpful for future studies and applications.
>
> - **Additional ablation studies**. We provided the ablation study on the shape of $\gamma_t$ in the Appendix D, although this analysis extends beyond the original scope of this paper.
>
>  **Lack of Reliability for the Proposed Evaluation Metric**
>
> - **Visualized images**. We presented  more visualized images generated by different models and compared their FID and AFD in Figure 5 and Appendix F.
> - **Validation**. We thoroughly validated the effectiveness of our proposed metric, AFD, for measuring conditional diversity and demonstrated its complementary role to FID, in Appendix A. To this end, we designed two classes of pseudo-generative models capable of controlling the diversity of the generated images. The performance of these models was further evaluated using both FID and AFD on the ImageNet dataset.
>
> **Questions on Lack of Theoretical Explanation**
>
> We added an illustration of the framework for constructing diffusion bridge models, shown in Fig. 1. The parameters $b$, $\gamma_t$, and $\epsilon_t$ govern the stochasticity introduced at three main stages: preprocessing, training, and sampling. Specifically, $b$ determines the noise added to the base distribution during preprocessing, $\gamma_t$ controls the noise introduced into the transition kernel, impacting both training and sampling, and $\epsilon_t$ regulates the noise added to the sampling SDEs, affecting only the sampling stage.
>
> **Questions on the purpose of Figure 1**
>
> We updated Figure 1 to illustrate our framework. Due to space constraints, we have moved it to the Appendix D and included additional explanations to provide more clarity.

---

> ### Comment · Reviewer_o3pD · 2024-11-22
> **Official Comment by Reviewer o3pD**
>
> Thank you for your response. The author’s numerical ablation studies have addressed my primary concerns. Therefore, I raise my score to 6.

---

### Official Review · Reviewer_Zibe · 2024-11-03

**Soundness:** 3
**Presentation:** 3
**Contribution:** 3
**Rating:** 6
**Confidence:** 1

**Summary:**

The authors introduce the Stochasticity-controlled Diffusion Bridge (SDB), a new theoretical framework that expands diffusion bridge design. SDB mitigates singularities during training and sampling, achieving speeds up to five times faster than the baseline while improving image quality and diversity by managing stochasticity in the transition kernel and base distribution.

Due to my limited familiarity with the field addressed in this paper, it is challenging for me to provide in-depth insights. However, I can offer some feedback from a more application-oriented perspective.

**Strengths:**

The Stochasticity-controlled Diffusion Bridge (SDB) demonstrates significant improvements in image quality, sampling efficiency, and conditional diversity, establishing new benchmarks in these aspects. Furthermore, the authors' introduction of score reparameterization and specially designed discretization effectively addresses challenges related to singularities.

**Weaknesses:**

SDS is a crucial loss function in the text-to-3D domain. Could the authors explore similar applications in this paper? It would be beneficial to demonstrate its extension in a 3D context.

**Questions:**

see weakness

---

> ### Author Response · Authors · 2024-11-19
> **Reply to reviewer Zibe**
>
> **SDS is a crucial loss function in the text-to-3D domain. Could the authors explore similar applications in this paper? It would be beneficial to demonstrate its extension in a 3D context.**
>
>
> Thank you for highlighting the significance of Score Distillation Sampling (SDS) in the text-to-3D domain and suggesting its exploration within the scope of our work. While our proposed sampling framework, which focuses on diffusion bridges and conditional diversity, is inherently flexible, exploring 3D applications is beyond the scope of this paper, which could be explored in future research.

---

> ### Comment · Reviewer_Zibe · 2024-11-26
>
> Thank you for your response ~

---

### Official Review · Reviewer_PqyK · 2024-11-04

**Soundness:** 3
**Presentation:** 3
**Contribution:** 2
**Rating:** 5
**Confidence:** 3

**Summary:**

This paper introduces the Stochasticity-Controlled Diffusion Bridge (SDB), a framework that enhances diffusion-based generative models by using a flexible, transition kernel-based approach. Unlike standard DDMs, which start from Gaussian noise, SDB can transition between arbitrary distributions, making it more versatile for tasks like image-to-image translation. Key innovations include a Stochasticity Control (SC) mechanism that regulates noise and drift terms, boosting stability, efficiency, and output diversity. Experiments show SDB operates 5× faster than DDBM and achieves an FID score of 0.89 with just five evaluations, setting a new quality benchmark.

**Strengths:**

1. The paper ensures the sampled distributions adhere to a mathematically sound process. Additionally, the proofs offered for Lemma 1 and Theorem 3 validate that the stochastic control can maintain desired marginal distributions across time

2. The method demonstrates strong performance in preserving alignment and detail within conditional sampling tasks. The model captures and retains critical details between the source and target distributions, resulting in high-fidelity outputs that stay true to the given conditions.

**Weaknesses:**

1.  The presentation of stochastic control in sections 3 and 4 mixes well-known stochastic process theory with the paper’s innovations, making it challenging to identify what is genuinely novel. Clarifying how the proposed SC mechanism specifically advances the field would help distinguish the paper’s contributions from established concepts.

2. The paper doesn’t clearly demonstrate why the proposed stochastic control outperforms other methods. While several design choices are introduced, their practical advantages and rationale are not sufficiently justified, leaving the reader uncertain about the improvements gained from these specific approaches.

3.The experiments appear simplistic and don’t convincingly support the claims, especially regarding generalization to real-world applications. Additionally, the diversity metric (Average Feature Distance) and image quality metrics could be better validated on more extensive benchmarks to strengthen the results.

**Questions:**

Is it possible to convert the sampling process into an ODE process by leveraging the Fokker-Planck equation?

---

> ### Author Response · Authors · 2024-11-19
> **Reply to reviewer PqyK**
>
> **Unclear innovations.**
>
> - We have moved the background material on stochastic process theory to the Appendix B, focusing the main text on our novel contributions. All theorems and formulas in Sections 3 and 4 are derived by us, ensuring that the originality of our work is clearly highlighted.
> - This work is the first to derive sampling SDEs for diffusion bridges conditioned on endpoints, specifically addressing the evolution of $q(X_t \mid x_T)$ for arbitrary Gaussian transition kernels. Unlike Stochastic Interpolants, which model the full data distribution $p_0$,  our framework directly addresses sampling from the conditional distribution $p_{0 \vert T}$. This novel focus simplifies both training and inference processes, representing a significant advancement in conditional generation techniques.
> - Our framework unifies and simplifies other mainstream diffusion bridge models, such as DBIM and I2SB. This generality demonstrates the versatility and practical impact of our approach.
> - Our approach is the first to highlight the issue of lacking conditional diversity in diffusion bridge models and to resolve it by introducing stochasticity into the base distribution.
> - Our proposed sampler is novel, achieving much better results than contemporaneous work DBIM.
> - Our score reparameterization is extension work, DDBM has the same reparameterization, we extend their work to fit any given translation kernel.
>
> **Questions on why the proposed stochastic control outperforms other methods.**
>
> - **Ablation studies on sampling algorithms, $\gamma_{\max}$ and $\epsilon_t$**. We added ablation studies on different sampling algorithms, $\gamma_{\max}$, and $\epsilon_t$, with results presented in Section 4 (Experiments) and illustrated in Figure 3.
>
> - **Analysis of training stability**. We added new experiments to analyze the training stability introduced by the proposed reparameterization. These results, which also provide insights into numerical stability, are shown in Figure 6 in Appendix D.
> Enhanced diversity evaluation. To strengthen diversity evaluation, we included DBIM in Table 4 for a direct comparison with our SDB model and other baselines.
>
> - **More design guidelines**. We added more design guidelines in the Appendix D. Although these experiments extend beyond the original scope of the paper, we have also added discussions on $\alpha_t$, $\beta_t$, and different shapes of $\gamma_t$ in Appendix D to provide a more comprehensive analysis.
>
> **Simplistic experiments, generalization of the model and validatation of AFD.**
>
> - **Generalization issue**. Our experiments are conducted on the Edges2handbags and DIODE datasets, which are widely recognized benchmarks for image-to-image translation tasks. These datasets were chosen because they are commonly used in the literature, making it easier to compare our results with other models and evaluate the effectiveness of our approach. It is important to note that EDM and I2SB—which are special cases of our proposed framework—have already been shown to perform effectively across a wide range of real-world applications. This indirectly demonstrates the generalization capabilities of our framework, as it encompasses these state-of-the-art methods while offering additional flexibility and enhancements. We acknowledge the importance of testing on larger-scale and more complex datasets to better reflect real-world scenarios. While this is beyond the scope of the current paper, in the future we will apply our model to more applications, e.g., brain imaging.
>
> - **Additional evaluation on more dataset and $\gamma_{\max}$.** We include additional experiments with $\gamma_{\max}=0.25$ and DIODE ($256 \times 256$) dataset, as shown in Table 3.
>
> - **Average Feature Distance**.  To further validate the robustness of our metrics, we included additional visual results to illustrate that a larger AFD corresponds to greater diversity in colors and textures, aligning with human judgment. We thoroughly validated the effectiveness of our proposed metric, AFD, for measuring conditional diversity and demonstrated its complementary role to FID, in Appendix A. To this end, we designed two classes of pseudo-generative models capable of controlling the diversity of the generated images. The performance of these models was further evaluated using both FID and AFD on the ImageNet dataset.
>
> **Questions on converting the sampling process into an ODE process.**
>
> Yes, it is. In our framework we can simply let $\epsilon_t = 0$ to convert the sampling SDE to ODE.

---

> ### Author Response · Authors · 2024-11-25
> **Follow up**
>
> Dear Reviewer PqyK:
>
> We greatly appreciate your careful reading and thoughtful feedback. Based on your suggestions, we have added clarification on our innovations, ablation studies, design guidelines and new diversity metric test. Please let us know if our responses fully addressed your concerns. We are always open to further discussion and welcome any suggestions to further improve our work!

---

> > ### Comment · Reviewer_PqyK · 2024-11-25
> > **Thank you for the clarification, but I still have concerns about the novelty.**
> >
> > Dear Authors,
> > Thank you for your active responses to my questions. I have thoroughly read your answers and the updated version. However, I still have concerns about the novelty of the points claimed in your method, which I believe is a variation of the existing theoretical framework. Additionally, since the experiments were conducted only on a small-scale dataset, I am also concerned about the real insights and impact of the proposed method.

---

> > > ### Author Response · Authors · 2024-11-25
> > > **Reply about the novelty**
> > >
> > > Dear Reviewer PqyK,
> > >
> > > We have discussed the comparisons with existing work in detail in our paper (Sec. 6 Related Work). If there is some other similar framework that you believe we have missed, please let us know what it is.
> > >
> > > The datasets that we have tested on are standard benchmarks, and are also the ones used in other similar published works [I2SB, DDBM], the edges2handbag($64\times64$) dataset contains 138567 images while DIODE($256\times256$) contains 16502 images.  Is there a particular dataset that you believe would enhance the understanding of our core contribution, the theoretical framework.
> > >
> > > [1] Liu G H, Vahdat A, Huang D A, et al. I $^ 2$ SB: Image-to-Image Schr\" odinger Bridge[J]. arXiv preprint arXiv:2302.05872, 2023.
> > >
> > > [2] Zhou L, Lou A, Khanna S, et al. Denoising diffusion bridge models[J]. arXiv preprint arXiv:2309.16948, 2023.

---

> > > > ### Comment · Reviewer_PqyK · 2024-11-25
> > > >
> > > > what is the key novel points of your framework compared to the following work where h-transform conditioned on endpoints and other conditions have been proposed
> > > > Liu, Xingchao, et al. "Let us build bridges: Understanding and extending diffusion generative models." arXiv preprint arXiv:2208.14699 (2022).
> > > >
> > > > Ye, Mao, Lemeng Wu, and Qiang Liu. "First hitting diffusion models for generating manifold, graph and categorical data." Advances in Neural Information Processing Systems 35 (2022): 27280-27292.
> > > >
> > > > what is the novel points of your framework about Schrödinger Bridge design space compared to the following work
> > > > Shi, Yuyang, et al. "Diffusion Schrödinger bridge matching." Advances in Neural Information Processing Systems 36 (2024).
> > > >
> > > > Is "specifically addressing the evolution of  for arbitrary Gaussian transition kernels" a vague and over claim.
> > > >
> > > > Since the paper claims to propose a new theoretical framework, I will not ask the reviewers to compare it with training-based and training-free methods on large pre-trained models, even though I am concerned about the real-world application impact of this approach compared to Yu, Jiwen, et al.'s 'Freedom: Training-free energy-guided conditional diffusion model' (Proceedings of the IEEE/CVF International Conference on Computer Vision, 2023). However, more visual results beyond just 'bags' should be presented, e.g. at lease some other category sketches extracted, e.g. Photo-Sketching:Inferring Contour Drawings from Images.

---

> ### Author Response · Authors · 2024-11-25
> **Reply to Reviewer PqyK**
>
> **Theoretic contribution**
>
> - Firstly, diffusion bridge matching framework [1], [2], [3] [4] build a pinned process $p(\cdot \mid x_0, x_T)$ using SDEs, then do bridge matching to find $p(X_t)$. However, in the I2I translation tasks, we want to build a link between Gaussian translation kernel $p(x_t \mid x_0, x_T) = \mathcal{N}(x_t; \mu(x_0, x_T), \gamma_t^2 I)$ and the conditional process $p(x_t \mid x_T)$. Our framework achieve this goal: Given transition kernel, we can find related sampling SDEs that related to $p(x_t \mid x_T)$. There are two difference compared to diffusion bridge matching framework: 1. we started from Gaussian transition kernel, 2. our sampling SDEs related to $p(x_t \mid x_T)$ instead of $p(x_t)$.
>
> - Secondly, to evolve sampling SDEs by adding score term, diffusion bridge matching and stochastic interpolants [5] framework need to train at least two models. While our framework only need to train one model to approximate $\mathbb{E}[x_0 \mid x_t, x_T]$, the score can be determined by score reparameterization.
>
> - Thirdly, diffusion Schrödinger bridge is different from I2I translation. We take diffusion Schrödinger bridge matching for example: 1. for image translation tasks, we do not have and care about the reference process, it's unnecessary to match reference process, which leads to alternative optimization for two objectives;  2. Diffusion Schrödinger bridge don't consider divergence free term, while I2I translation should consider. 3. Markovian projection in DSBM is to approximate $\mathbb{Q}$ given $\mathbb{Q}(\cdot \mid x_0, x_T)$, while ours framework is to approximate $\mathbb{Q}(\cdot \mid x_T)$ given $\mathbb{Q} (x_t \mid x_0, x_T)$.
>
> **Is "specifically addressing the evolution of for arbitrary Gaussian transition kernels" a vague and over claim.**
>
> - Our framework gives sampling SDEs for any Gaussian translation kernel with the form $\mathcal{N}(x_t; \alpha_t x_0 + \beta_t x_T, \gamma_t^2I)$, it's better to say "specifically addressing the evolution of $q(X_t \mid x_t)$ for arbitrary Gaussian transition kernels with the form $\mathcal{N}(x_t; \alpha_t x_0 + \beta_t x_T, \gamma_t^2I)$" , but it should be good.
>
> **Dataset and visualization**
>
> For datasets and visualization, we also visualized depth image to real images with size $256 \times 256$ in the Appendix.
>
> [1] Liu, Xingchao, et al. "Let us build bridges: Understanding and extending diffusion generative models." arXiv preprint arXiv:2208.14699 (2022).
> [2] Peluchetti S. Non-denoising forward-time diffusions[J]. arXiv preprint arXiv:2312.14589, 2023.
> [3] Shi Y, De Bortoli V, Campbell A, et al. Diffusion Schrödinger bridge matching[J]. Advances in Neural Information Processing Systems, 2024, 36.
>
> [4] Ye, Mao, Lemeng Wu, and Qiang Liu. "First hitting diffusion models for generating manifold, graph and categorical data." Advances in Neural Information Processing Systems 35 (2022): 27280-27292.
>
> [5] Albergo M S, Boffi N M, Vanden-Eijnden E. Stochastic interpolants: A unifying framework for flows and diffusions[J]. arXiv preprint arXiv:2303.08797, 2023.

---

> > ### Comment · Reviewer_PqyK · 2024-11-25
> >
> > Thanks for the response, I will keep my rating of 5.

---

### Official Review · Reviewer_1o8f · 2024-11-04

**Soundness:** 2
**Presentation:** 3
**Contribution:** 3
**Rating:** 6
**Confidence:** 3

**Summary:**

This paper introduces Stochasticity-controlled Diffusion Bridge (SDB), a generalized diffusion bridge-based model framework that enables flexible control of stochasticity. Specifically, SDB considers stochasticity in 1) the transition kernel, 2) sampling SDE, and 3) the base distribution. This is achieved by designing a noise schedule for the transition kernel, regulating the drift term in the sampling SDEs, and adding noise into the base distribution. The paper also proposes a score reparameterization and a discretization sampling scheme to mitigate singularity. Experiments show that the proposed SDB has better image quality, efficiency, and diversity than other baselines in the image-to-image translation task.

**Strengths:**

1. The proposed SDB framework provides a more generalized view of diffusion bridge-based models (DBM). Under the framework, this work highlights the importance of randomness in DBM. It shows that carefully selected randomness in sampling SDEs, transition kernel, and the base distribution can significantly improve sampling efficiency, image quality, and diversity.

2. The proposed score reparameterization and discretization sampling scheme by design is more likely to avoid singularity and instability in training and sampling.

3. In experiments, the proposed method SDB notably outperforms previous DBM methods in quality, efficiency, and diversity, as evaluated by several quantitative metrics.

**Weaknesses:**

1. The main issue of this paper is insufficient evaluation. First, the ablation study is not thorough enough. It is not shown how different choices of $\gamma_{max}$ affect the results. In addition, there's no ablation study to show the effects of the proposed reparameterization and sampling algorithm. Thus, it is unclear how much gain each technique contributes to the final model.
Regarding the verification of reparameterization, apart from the normal metrics, I also suggest analyzing the numerical stability during training/sampling to show that the proposed technique indeed mitigates singularity.

2. Qualitative comparison is insufficient. In the main paper, only Fig. 4 shows the qualitative comparison between SDB and DDBM. However, a comparison to other methods such as DBIM should also be provided. I also suggest including DBIM in Table 4 so that the SDB is compared to more than one baseline in terms of diversity.

3. The randomness added to the base distribution lacks further explanation. Why convolving $\pi_{cond}$ with a Gaussian rather than adding Gaussian noise to $\pi_{cond}$? What's the kernel size for the Gaussian?

4. DBIM is not discussed in related works.

Typo: Line 212 ODE -> SDE.

**Questions:**

The settings of Table 2 and Table 3 are not matched. For example, Table 2 shows that $\eta = 0.3$ is the best, but Table 3 uses $\eta = 1$. The quantitative scores are also not matched for $\eta = 1$ on edges2handbags. Please explain the inconsistency.

---

> ### Author Response · Authors · 2024-11-19
> **Reply to reviewer 1o8f**
>
> **Insufficient Evaluation**
>
> - **Ablation Studies on sampling algorithms, $\gamma_{\max}$ and $\epsilon_t$**: we added ablation studies on different sampling algorithms, $\gamma_{\max}$, and $\epsilon_t$. These results are now presented in Section 4 (Experiments), as shown in Figure 3.
> - **Design guidelines on $\alpha_t$, $\beta_t$ and the shape of $\gamma_t$**. Although the design of $\alpha_t$, $\beta_t$ and the shape of $\gamma_t$ beyond the original scope of the paper, we have also added discussions on design guidelines on $\alpha_t$, $\beta_t$, and ablation study on different shapes of $\gamma_t$. See Appendix D for more details.
> - **Verification of Reparameterization**. In Table 4, we demonstrate that reparameterizations used in DDBM and EDM can be incorporated into our framework. Their experimental results and discussions further support our arguments. Additionally, we include new experiments analyzing the training stability brought by the proposed reparameterization. These results, including insights on numerical stability, are shown in Figure 7 in Appendix D. This further substantiates our claim that the proposed technique mitigates singularity issues effectively.
> - **Validation of AFD**. We thoroughly validated the effectiveness of our proposed metric, AFD, for measuring conditional diversity and demonstrated its complementary role to FID, in Appendix A. To this end, we designed two classes of pseudo-generative models capable of controlling the diversity of the generated images. The performance of these models was further evaluated using both FID and AFD on the ImageNet dataset.
>
>
> **Insufficient qualitative comparison.**
>
> - **Additional evaluation on more dataset and $\gamma_{\max}$**. We include additional experiments with $\gamma_{\max}=0.25$ and DIODE ($256 \times 256$) dataset, as shown in Table 3.
> - **Adding baseline DBIM in Fig. 5 and Table 4**. While DBIM is a contemporaneous work and its official code has not been released, we note that its sampling algorithm can be incorporated into our framework. To address this, we implemented DBIM within our framework and conducted comparisons. These results, including experiments with different samplers, $b$, and NFEs, provide further evidence supporting our conclusions. To enhance the diversity evaluation, we have added DBIM to Table 4 for direct comparison against our SDB model and other baselines. In Figure 5, we added a qualitative comparison of DDIM, which illustrates the advantages of our approach.
>
> **Question on randomness added to the base distribution.**
>
> The convolution of $\pi_{\text{cond}}$ with a Gaussian is equivalent to adding Gaussian noise to the samples. This operation ensures that the randomness is introduced in a way consistent with the probabilistic framework.
>
> **DBIM is not discussed in related works.**
>
> Thank you very much for your reminder, we added DBIM in our related works.
>
> **Typo: Line 212 ODE -> SDE.**
>
> Thank you very much for your reminder, the typo has been corrected.
>
> **Questions on the setting of $\eta$ in Table 2 and 3.**
>
> Thank you very much for your question. Results in Table 2 are based on the DDBM pre-trained model using our proposed sampling algorithm. The optimal $\eta$ for DDBM under our evaluation was found to be 0.3. Results based on our SDB model are based on our SDB model, which achieved the best performance at $\eta=1.0$. To address the reviewer's concern, we have conducted an ablation study for $\eta$ in the context of the SDB model, shown in Fig. 3. Additionally, we have added explanations in the main text to clarify the differences.

---

> > ### Comment · Reviewer_1o8f · 2024-11-26
> > **Official Comment by Reviewer 1o8f**
> >
> > Thank you for your response, which has addressed most of my concerns. I have two follow up questions though:
> > 1. In Figure 7, the training curve of "without score reparameterization" is more stable than "with score reparameterization". This is in contrast with the claim that the score reparameterization stabilize training. Why is that?
> > 2. In Figure 5, should DDIM be DBIM? My understanding is that DDIM does not do image translation task.

---

> > > ### Author Response · Authors · 2024-11-26
> > > **Reply to Reviewer 1o8f**
> > >
> > > Thank you very much for your careful reading and for pointing out these two typos. We have corrected them. In Figure 7, the more stable curve is now labeled as "with score reparameterization." Additionally, in Figure 5, the correct term is "DBIM."

---

> > > > ### Comment · Reviewer_1o8f · 2024-11-27
> > > > **Official Comment by Reviewer 1o8f**
> > > >
> > > > Thank you for your response. I have raised my score to 6.

---

> ### Author Response · Authors · 2024-11-25
> **Follow up**
>
> Dear Reviewer 1o8f;
>
> We greatly appreciate your careful reading and thoughtful feedback. Based on your suggestions, we have added additional ablation studies and qualitative comparison, we also added more discussion about inovations, design guidance and metric test. Please let us know if our responses fully addressed your concerns. We are always open to further discussion and welcome any suggestions to further improve our work!

---

### Author Response · Authors · 2024-11-21
**Official Comment by Authors**

We thank the reviewers for their comments. Although reviewers agreed that our more generalized framework highlights the importance of randomness in diffusion bridge models, demonstrating superior performance in FID, sampling efficiency, and conditional diversity on I2I tasks, there were questions about our inovations, how different aspects of our method contribute to the strength of the results, and about the new metric we introduced to measure conditional diversity.

 We summarize here the main points addressed in responses to individual reviewers.

**1. Clarified inovations**.

- **Streamlined focus**. Background on stochastic processes moved to Appendix B. All theorems and formulas in Sections 3 and 4 are original, emphasizing the uniqueness of our work.
- **Framework clarity**. Added an illustration in Fig. 1 and clarified theoretical contributions in the conclusion.
- **Refined theoretical contributions**. We clafied our theoretical contributions in the Conclusion.  This is the first study to derive sampling SDEs for diffusion bridges conditioned on endpoints, addressing the evolution of  $q(X_t \mid x_T)$  for arbitrary Gaussian transition kernels. Unlike Stochastic Interpolants (Albergo et al., 2023), which model the full data distribution $p_0$, our framework directly targets sampling from the conditional distribution $p_{0 \mid T}$, simplifying both training and inference and marking a significant step forward in conditional generation techniques.
- **New insight to denoising diffusion models**. Our framework can also bring new insight for denosing diffusion models. We note that [TrigFlow (Lu & Song, 2024)](https://openreview.net/forum?id=LyJi5ugyJx), a main contribution of a contemporaneous submission, employs the same score reparameterization and pre-conditioning techniques. Notably, TrigFlow can be considered a special case of our framework by setting $\alpha_t = \cos(t)$, $\beta_t=0$, $\gamma_t=\sigma_0 \sin(t)$, $t\in[0,\frac{\pi}{2}]$. This further demonstrates the validity, generalization and forward-looking nature of our theoretical framework. We included this discussion in Appendix E to further strengthen our theoretical contribution.



**2. New experiments and design guidelines**.

- **Ablation studies on sampling algorithms, $\gamma_{\max}$, and $\epsilon_t$**. We added ablation studies on different sampling algorithms, $\gamma_{\max}$, and $\epsilon_t$. These results are now presented in Section 4 (Experiments), as shown in Figure 3.
- **Additional evaluation on more dataset and $\gamma_{\max}$**. We include additional experiments with $\gamma_{\max}=0.25$ and DIODE ($256 \times 256$) dataset, as shown in Table 3.
- **Additional baseline**.  While DBIM (Zheng et al., 2024) is a contemporaneous work and its official code has not been released, we note that its sampling algorithm can be incorporated into our framework. To enhance the diversity evaluation, we have added DBIM to Table 4 for direct comparison against our SDB model and other baselines. In Figure 5, we added a qualitative comparison of DDIM, which illustrates the advantages of our approach.
- **Verification of Reparameterization**. We include new experiments analyzing the training stability brought by the proposed reparameterization. These results, including insights on numerical stability, are shown in Figure 6 in Appendix D.
- **Enhanced diversity evaluation**. In Table 4, we added more evaluation on different samplers and different choices of $b$.
- **Design guidelines on $\alpha_t$, $\beta_t$ and the shape of $\gamma_t$**. Although the design of $\alpha_t$, $\beta_t$ and the shape of $\gamma_t$ beyond the original scope of the paper, we have also added discussions on design guidelines on $\alpha_t$, $\beta_t$, and ablation study on different shapes of $\gamma_t$. See Appendix D for more details.

**3. New diversity metric test**.

- **Visualized images**. We presented more visualized images generated by different models and compared their FID and AFD in Figure 5 and Appendix F.
- **Validation**. We added new experiments to validate that our diversity metric correctly captures diversity in a scenario where we have ground truth, in Appendix A. To this end, we designed two classes of pseudo-generative models capable of controlling the diversity of the generated images. The performance of these models was further evaluated using both FID and AFD on the ImageNet dataset.



[1]. Michael S Albergo, Nicholas M Boffi, and Eric Vanden-Eijnden. Stochastic interpolants: A unifying framework for flows and diffusions. arXiv preprint arXiv:2303.08797, 2023.

[2]. Kaiwen Zheng, Guande He, Jianfei Chen, Fan Bao, and Jun Zhu. Diffusion bridge implicit models.
arXiv preprint arXiv:2405.15885, 2024.

[3]. Cheng Lu and Yang Song. Simplifying, stabilizing and scaling continuous-time consistency models.
arXiv preprint arXiv:2410.11081, 2024.

---

### Meta-Review · Area_Chair_PQi5 · 2024-12-17

**Metareview:**

This paper proposes a novel theoretical framework, Stochasticity-controlled Diffusion Bridge (SDB), to enhance the performance of diffusion bridge models (DBMs) in image-to-image (I2I) translation tasks. The authors claim that by meticulously controlling the stochasticity in sampling SDEs, transition kernels, and base distributions, they can significantly improve sampling efficiency, image quality, and diversity. They further assert that their framework offers strategies to mitigate singularities during both training and sampling.   The authors report achieving up to 5x faster sampling speeds compared to baseline models, along with lower FID scores, indicating improved image quality. They also claim to achieve new benchmarks in image quality and sampling efficiency by managing stochasticity within the transition kernel. Additionally, they introduce a new metric to quantify image diversity and demonstrate that introducing stochasticity into the base distribution significantly improves this metric.

Strengths:
- The paper presents a novel perspective on controlling stochasticity in diffusion bridge models for I2I translation. The proposed SDB framework introduces innovative mechanisms to manage noise levels in different components of the model.
- The paper demonstrates strong empirical results, showing improvements in terms of both FID scores and sampling efficiency on standard (small scale) datasets. The proposed method reportedly achieves state-of-the-art results on standard I2I translation benchmarks.
- The paper provides a solid theoretical grounding for the proposed SDB framework.

Weaknesses:
- Limited Practical Impact: Concerns were raised about the limited scope of the experiments and the lack of evidence demonstrating the impact of the proposed method on real-world applications. The focus on theoretical contributions may overshadow the practical implications of the work, but the theoretical contribution per-se don't seem strong enough.
- Clarity and Novelty: Despite revisions, concerns remain about the novelty of the proposed framework, with some reviewers suggesting it might be a technical variation of existing results rather than a genuinely new approach.
- Experimental Validation: While the paper presents empirical results, questions were raised about the sufficiency of the experimental validation, particularly the choice of datasets and the depth of the ablation studies.

Despite the potential of the SDB framework and the reported performance improvements, concerns about the novelty, practical impact, and experimental validation of the work persist. The reviewers' doubts about the core contribution's novelty and the limited evidence of its effectiveness in real-world scenarios.

**Additional Comments On Reviewer Discussion:**

During the rebuttal period, reviewers provided constructive feedback, leading to several revisions and clarifications in the paper. Key points raised included concerns about the novelty of the approach, the clarity of the presentation, and the level of experimental validation. The authors addressed these concerns by clarifying their innovations, streamlining the focus, and adding new experiments and design guidelines.

The reviewers seem to agree that the paper is technically sound and well-written. The main point of contention is the novelty of the proposed framework. While Reviewer o3pD sees it as promising, Reviewers 1o8f and PqyK have reservations. Reviewer PqyK also raises concerns about the balance between theory and practical application, suggesting the paper might be overselling its theoretical contributions.

The authors need to clearly articulate the novelty and significance of their framework, distinguishing it from existing work like [Sarkaa2019]. They should also consider strengthening the discussion of the real-world impact and applications to address Reviewer PqyK's concerns.

The inconsistencies and inaccuracies in one of the reviews were also taken into account in the final decision.

[Sarkaa2019] Särkkä, S. and Solin, A., 2019. Applied stochastic differential equations (Vol. 10). Cambridge University Press.

---

### Decision · Program_Chairs · 2025-01-22

Reject